# Temporal Enhancement of Contrastive Audio–Language Models through Self–Supervised Post–Training with Text-Audio Pairs

## Abstract

Research on multi-modal contrastive learning strategies for audio and text has rapidly gained interest. Contrastively trained Audio-Language Models (ALMs), such as CLAP, which establish a unified representation across audio and language modalities, have enhanced the efficacy in various subsequent tasks by providing good text aligned audio encoders and vice versa. These improvements are evident in areas like zero-shot audio classification and audio retrieval, among others. However, the ability of these models to understand natural language and temporal relations is still a largely unexplored and open field for research. In this paper, we propose to equip the multi-modal ALMs with temporal understanding without loosing their inherent prior capabilities of audio-language tasks with a temporal instillation method **TeminAL**. We implement a two-stage training scheme TeminAL A & B, where the model first learns to differentiate between multiple sounds in TeminAL A, followed by a phase that instills a sense of time, thereby enhancing its temporal understanding in TeminAL B. This approach results in an average performance gain of $5.28\%$ in temporal understanding on the benchmark ESC-50 dataset, while the model remains competitive in zero-shot retrieval and classification tasks on the AudioCap/Clotho datasets. We also note the lack of proper evaluation techniques for contrastive ALMs and propose a strategy for evaluating ALMs in zero-shot settings. The general-purpose Zero-Shot Temporal Evaluation **(ZSTE)** strategy , is used to evaluate various prior models. ZSTE demonstrates a general strategy to evaluate all ZS contrastive models. The model trained with TeminAL successfully outperforms current models on most downstream tasks.

## 1 Introduction

Audio, text, and images are among the most prevalent forms of information data. Developing models with multi-modal capabilities is well recognized as a path forward toward artificial general intelligence (Fei et al., 2022; Huang et al., 2021). In the field of multi-modal learning, contrastive learning has emerged as an effective strategy for training models on extensive, less-structured internet-sourced data (Radford et al., 2021; Liang et al., 2022; Tian et al., 2020). Contrastive learning-based models have demonstrated exceptional adaptability across a range of related tasks, such as image classification (Chen et al., 2020; He et al., 2020a), natural language processing (Gao et al., 2021) and speech processing (Ravanelli et al., 2020), making them a crucial area of research in multi-modal machine learning. One notable early model in this domain is CLIP, developed by Radford et al. (2021). CLIP learns the relationship between text and images, aligning them in a common latent domain. It stands out as a groundbreaking vision-language model, enabling tasks such as generating images from text (Rombach et al., 2022) and formulating image captions (Mokady et al., 2021).

Similar work on contrastive learning has been extended to other multi-modal domains, such as video-language (Xu et al., 2021; Fang et al., 2021; Zhao et al., 2022; Luo et al., 2022; Cheng et al., 2023; Ge et al., 2022) and audio-language models (Elizalde et al., 2023; Huang et al., 2022; Guzhov et al., 2022; Wu et al., 2023b; Deshmukh et al., 2023; Wu et al., 2023a). Contrastive models generally excel in relating different modalities through their learned embedding and performing similarity-based

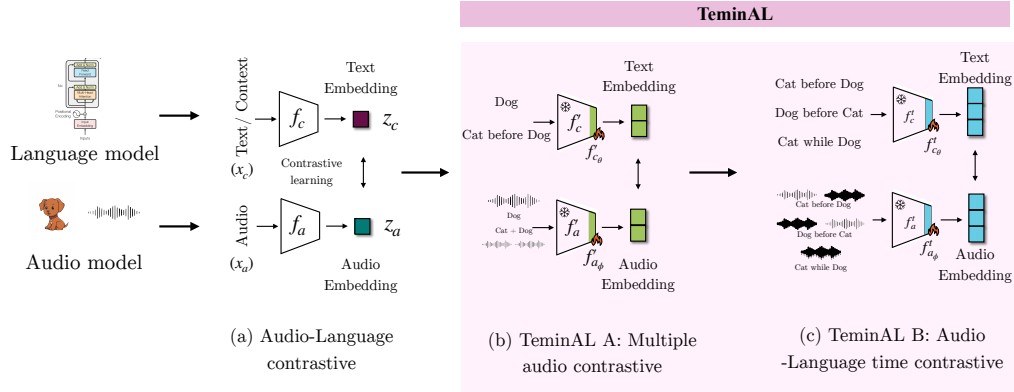

Figure 1: The overview of TeminAL where we are post–training orginal CLAP encoders $f_c$ and $f_a$ with our **TeminAL** method to get $f_c^t$ and $f_a^t$ after application of the two–stage training. We only train a subset of the total weights ($f_{c_\theta}^t$ and $f_{a_\phi}^t$) in both our training stages. Mathematical formualtion of the functions are elaborated in section 3.3 and section 3.4.

retrieval tasks. These multi-context encoders integrate well with other downstream models, such as retrieval and open-ended generation models (Ramesh et al., 2021; Li et al., 2022; Yuan et al., 2021; Singh et al., 2022). However, previous authors have shown the limitations of audio-language models in truly understanding natural language while learning the relationship between texts and audio (Wu et al., 2023a; Ghosh et al., 2023). Critical applications like medical procedures, assembly instructions, commercial user applications, cooking instructions, and language learning may suffer from mistaken outputs in either text or audio settings. Wu et al. (2023a) highlights a critical limitation in current audio-language models (ALMs): a bias towards retrieving nouns and verbs, often at the expense of understanding the complete sentence context. They illustrate this by training an ALM on captions stripped of all but nouns and verbs, achieving performance comparable to or even surpassing models trained on full, non-shuffled captions. This finding questions the prevailing assumption that ALMs require holistic sentence comprehension for high performance, revealing gaps in their compositional reasoning capabilities. Furthermore, studies such as Thrush et al. (2022), Ma et al. (2023), and Yuksekgonul et al. (2022) have demonstrated that models like CLIP struggle with language reasoning despite access to extensive training datasets. These limitations arise because contrastive pre-training primarily emphasizes retrieval tasks, enabling strong benchmark performance without a deep understanding of sentence composition. In response to these challenges, Ghosh et al. (2023) critique existing audio-retrieval benchmarks, arguing that the perceived success of ALMs often lacks true compositional understanding. They introduce CompA-CLAP, an ALM designed with novel contrastive training techniques to improve both language comprehension and attribution capabilities in multiple training steps but with the same global objective of making the model directly adapt to temporality. Although the model perform well on various downstream tasks, these approaches do not adequately address a fundamental prerequisite for compositional reasoning in audio tasks: the ability to distinguish multiple sound events before attempting to establish relationships between them. Our work emphasizes this overlooked step, proposing a framework where the model first learns to recognize the existence of multiple sound events as a foundation for higher-level reasoning. Similarly Yuan et al. (2024); Wu et al. (2023a) trains a contrastive learning model without requiring to address the need of multiple sounds distinction which defeats the purpose of increasing the interpretability of the models.

In contrast, our approach achieves this advancements within a limited computational budget, training around 10% of the total trainable parameters of the base model (here CLAP) and utilizing a single dataset (ESC-50). Unlike prior works, such as those by (Ghosh et al., 2023; Yuan et al., 2024; Wu et al., 2023a), which rely on more expansive datasets and substantial computational resources. Our focus is on developing a methodology that can effectively instill a sense of time in the model within acceptable computational constraints, rather than on generalizing over large, diverse datasets. Our approach detailed in section 3 and illustrated in fig. 1, modifies the contrastive training

paradigm by introducing a multi-stage hierarchical training process. In the first stage, the model is trained to recognize and differentiate multiple sound events. In the subsequent stage, it learns the temporal relationships between these events, addressing limitations of prior contrastive models that focus solely on text-audio pair similarities without incorporating temporal language dynamics. The training objective is based on previous works of Oord et al. (2018) on the formulation of InfoNCE loss and Bagad et al. (2023) who explored temporal instillation in video-language models, however we take the research forward and implement a structured, multi-step post-training process tailored to complex temporal tasks in the audio-language domain. Our objective Comparative analysis in section 5 demonstrates the necessity and efficacy of this approach, showing that our two-stage process outperforms single-stage methods in enabling ALMs to comprehend audio-language modality relationships. This work establishes a significant advancement in ALMs by addressing foundational gaps in sound event distinction and temporal reasoning which has been overlooked in the past.

We further critique current zero-shot evaluation methods, which predominantly rely on basic similarity-based retrieval accuracies or employ large language models (LLMs) as evaluators, both of which have shown inherent biases and limitations (Gao et al., 2024; Jones & Steinhardt, 2023; Stureborg et al., 2024; Wang et al., 2023). Although previous models have been evaluated for their robustness over time (Shocher et al., 2018; Bau et al., 2019; Kundu et al., 2020; Huang et al., 2020; Sun et al., 2020; Liu et al., 2021), these assessments fail to test the models' general language and temporal understanding comprehensively. To bridge this gap, we propose a sequential zero-shot evaluation method that poses increasingly complex tasks, aiming to create a general-purpose evaluation framework (details discussed in algorithm 2).

**Main contributions.** Here are the key contributions of our work, which, to the best of our knowledge, are novel and not present in current state-of-the-art models:

- Our analysis indicates that current contrastive ALMs face challenges in accurately capturing temporal relationships between audio and text, as shown in table 3, highlighting an area for potential improvement in existing models.

- We propose a two step post–training within limited compute budget scheme **TeminAL**: **Tem**poral **In**stillation in **A**udio-**L**anguage Models for multi-modal contrastive ALMs. Aimed towards developing temporally aware contrastive audio & text encoders which can be employed in various close and open ended generation models as described in section 3.4.

- We propose **ZSTE**: **Z**ero **S**hot **T**emporal **E**valuation scheme for contrastively trained models. The sequentially complicated evaluation strategy used for evaluating our objectives of temporal instillation section 4.2.

## 2 BACKGROUND AND RELATED WORK

### 2.1 FOUNDATION MODELS AND MULTI-MODAL TEXT-AUDIO LEARNING

The expansion of Pretrained Foundation Models (PFMs) now includes auditory (Baevski et al., 2020), visual (Dosovitskiy et al., 2020), text-image (Ramesh et al., 2021; Radford et al., 2021), and multi-modal data (Lu et al., 2019; Akbari et al., 2021), driving multi-modal integration. Recent work uses audio-visual contrasts for sound localization (Chen et al., 2021; Wu et al., 2022a), cross-modal retrieval (Surís et al., 2022), and zero-shot classification (Wu et al., 2022b; Guzhov et al., 2022). Audio-text models are gaining traction, including those in the DCASE competition for audio retrieval with language (Xie et al., 2022), and PFMs have been applied in music tagging (Manco et al., 2022), environmental sound identification (Zhao et al., 2021; Lou et al., 2022; Mei et al., 2022; Koepke et al., 2022), and zero-shot tasks (Zhao et al., 2021; Lou et al., 2022; Mei et al., 2022; Koepke et al., 2022; Elizalde et al., 2023). Open-ended models (Kong et al., 2024; Chu et al., 2023; Liu et al., 2024; Deshmukh et al., 2023) enable QA capabilities, but our focus is on contrastive learning for audio encoders. The trend is towards integrating language into auditory systems, with applications in text-to-audio (Ghosal et al., 2023; Liu et al., 2023a; Huang et al., 2023), music generation from text (Agostinelli et al., 2023), and sound source separation (Liu et al., 2023b). Frameworks like CLAP and Compa (Elizalde et al., 2023; Ghosh et al., 2023) unify auditory-linguistic domains, offering strong zero-shot performance in multimodal tasks.

## 2.2 SELF-SUPERVISED LEARNING AND POST-TRAINING

**Self-Supervised Learning (SSL)** has revolutionized machine learning, especially in NLP and computer vision (He et al., 2020b; Bao et al., 2021). SSL involves training models to predict parts of their input using other parts, leveraging the data's inherent structure for supervision. A prominent SSL method, **Contrastive Learning**, learns representations by contrasting positive and negative examples, effectively distinguishing similar and dissimilar data samples (Radford et al., 2021; Liang et al., 2022; Tian et al., 2020; Chen et al., 2020; He et al., 2020a). This approach has significantly advanced representation learning, achieving state-of-the-art results across various domains (Chen et al., 2020; He et al., 2020a).

**Post-training** introduces an additional self-supervised phase to existing models using a limited set of data before downstream task evaluation, reducing the costs of initial large-scale training (Luo et al., 2022; Xue et al., 2022). Luo et al. (2022) employs static mean-pooling, whereas Xue et al. (2022) aligns image captions with video subtitles. In this unsupervised setting, post-training usually fine-tunes few parameters, maintaining the core strengths of the parent model.

## 2.3 ZERO-SHOT INFERENCE: LIMITATIONS OF CLASSICAL ZERO-SHOT RETRIEVAL

Zero-shot inference enables models to recognize unseen classes without relying on labeled data from each target class, unlike traditional supervised learning (Xian et al., 2018; Wang et al., 2020b). While zero-shot learning facilitates generalization to unseen classes, conventional audio-retrieval benchmarks often lack compositional complexity, typically involving single acoustic events without proper word order (Radford et al., 2021; Baevski et al., 2020; Gemmeke et al., 2017). In traditional audio classification, models are trained on specific classes like musical genres or environmental sounds, but zero-shot audio classification requires identifying audio samples from previously unseen classes. For example, a model trained on animal and vehicle sounds should also classify new categories like "machinery" or "insects" (Wang et al., 2020a). As illustrated in fig. 8, zero-shot classification involves encoding audio and text prompts through respective encoders and using cosine similarity to predict classes (Harwath & Glass, 2015; Kim & Pardo, 2018). Zero-shot audio retrieval extends this concept by finding relevant audio clips from unseen classes based on queries, such as retrieving "birdsong" or "ocean waves" when trained only on spoken words and ambient sounds (Fonseca et al., 2021). As shown in fig. 9, the process involves encoding prompts and audio clips, with cosine similarity determining the most relevant match (Chang & Yang, 2019). This approach leverages class information to understand semantic relationships.

# 3 METHODOLOGY

## 3.1 PRELIMINARIES

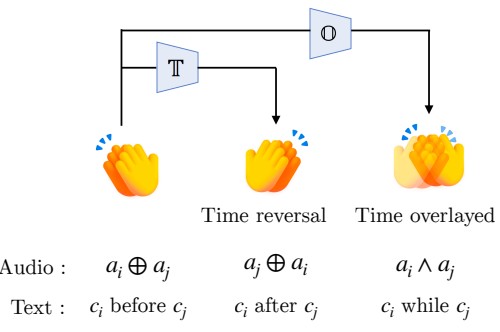

Audio :  $a_i \oplus a_j$      $a_j \oplus a_i$      $a_i \wedge a_j$

Text :  $c_i$ before $c_j$      $c_i$ after $c_j$      $c_i$ while $c_j$

Figure 2: Temporal Augmentations

**Introduction to Fundamentals.** Consider set $\mathbb{A}$ as the domain of audio recordings and $\mathbb{C}$ as the set of corresponding textual transcripts (contexts). For any two discrete and non-overlapping audio clips $\{a_i, a_j\}$ within $\mathbb{A}$, let their relevant transcripts be $\{c_i, c_j\}$ in $\mathbb{C}$ (We use '$c$' for transcripts to avoid confusion with the time variable '$t$'). We define an integrated segment that respects the sequential order as $(a_{ij}, c_{ij})$, with $a_{ij}$ constructed by the operation $[a_i \oplus a_j]$, which concatenates the two audio clips as marked by the operator $\oplus$ which shows the concatenation operation also shown in fig. 2. Similarly for contexts, we first introduce $\tau = \{\tau_t, \tau_o\}$ to represent a sequential relationship, where $\tau_t$ can either be preceding or succeeding as prompted by {*before* or *after*} and we define $\tau_o$ for overlapping language prompt {*while*}. Following which $c_{ij}$ is represented as $[c_i : \tau_t; c_j]$, merging the transcripts in a manner that it reflects the temporal relation $\tau = \{\tau_t, \tau_o\}$. Later in section 3.2, we relate $[a_i \oplus a_j]$ with $[a_j \oplus a_i]$

using mathematical operators. It should be noted that the arrangement of $a_i$ and $a_j$ within $a_{ij}$ may vary depending on the value of $\tau_t$. The same is applicable for overlapping sounds $(a_j, a_j)$, for which the overlapping texts can be represented by $[c_i : \tau_o; c_j]$, which essentially means "$c_i$ *while* $c_j$" with overlaid audios $[a_i \wedge a_j]$. For simplicity, we will refer to the composite audio-text pair $(a_{ij}, c_{ij})$ as $(a, c)$, except where additional specificity is required.

## 3.2 DATA-PROCESSING: DESIGNING OUR TRAINING DATA.

The dataset for our post–training study was meticulously curated from publicly available audio-text pairs, we specifically select the ESC-50 dataset for the current study. The dataset selection and processing is descried in detail in appendix B.2. We introduce a temporal inversion operator '$\mathbb{T}$' and temporal overlay operator '$\mathbb{O}$' to represent the transformation of audio and text training data to form the temporally inverted samples and temporally overlapped samples as shown in equation 1 for the temporal inversion and equation 2 for temporally overplayed samples. This function is designed to operate on pairs of simultaneous audios $(a_i, a_j)$ or transcription sequences $(c_i, c_j)$ where sequences in both these sets are initially non–overlapping. We show temporal addition/ concatenation of the pair of audios by $a_j \oplus a_i$ and overlaying of the audio pair by $a_j \wedge a_i$. Meanwhile temporal addition and overlaying of texts are shown as $c_j; \tau_t; c_i$ and $c_j; \tau_o; c_i$ respectively and follows the same convention as mentioned in section 3.1.

$$\mathbb{T}(a) = \mathbb{T}([a_i; a_j]) := [a_j \oplus a_i], \quad \mathbb{T}(c) = \mathbb{T}([c_i; c_j]) := [c_j; \tau_t; c_i] \tag{1}$$

$$\mathbb{O}(a) = \mathbb{O}([a_i; a_j]) := [a_j \wedge a_i], \quad \mathbb{O}(c) = \mathbb{O}([c_i; c_j]) := [c_j; \tau_o; c_i] \tag{2}$$

It is essential to recognize that '$\mathbb{T}$' does not literally reverse time within the audio tracks, rather it rearranges the sequence of events within the compiled segments. Our goal is to cultivate a model capable of distinguishing an original audio-text pair $(a, c)$ from both of its temporally inverted counterpart $(a, \mathbb{T}(c))$, and also $(\mathbb{T}(a), c)$; furthermore to contrast all of these from the overlaid text-audio pair as $(\mathbb{O}(a), c)$ (which is the same as $(a, \mathbb{O}(c))$). So a typical training batch would look like $B_{a_B} = \{a, \mathbb{T}(a), \mathbb{O}(a)\}$ for the audio and $B_{t_B} = \{c, \mathbb{T}(c), \mathbb{O}(c)\}$ for the text. The details for out data–preparation method is described in algorithm 3. As described earlier in section 1 we have a hierarchical 2–stage training process TeminAL A followed by TeminAL B. The text–audio dataset $\{B_{a_B}, B_{t_B}\}$ is used to train TeminAL B. While the first pretraining TeminAL A, works on learn single sounds and multiple sounds thus the input data in the batch doesn't consists of time–reversed data, it's made up of $B_{a_A} = \{a_i, a_i \oplus a_j \; \forall i, j \in \{1, N\}$ the audio and $B_{t_A} = \{c_i, c_i \oplus c_j \; \forall i, j \in \{1, N\}$ for the text.

## 3.3 PRELIMINARIES OF POST–TRAINING WITH SSL

The input texts and audios are first transformed into machine-level embeddings. Let the processed embedding for audio be $\boldsymbol{x}_a$ where $\boldsymbol{x}_a \in \mathbb{R}^{F \times T}$, with $F$ representing frequency components (e.g., Mel frequency bins) and $T$ indicating the number of temporal segments. The corresponding textual data is denoted as $\boldsymbol{x}_c$ for a given sample. For a batch of $N$ text-audio pairs, the audio and corresponding text are represented as $\{X_a, X_c\}_i = \{\boldsymbol{x}_a^{(i)}, \boldsymbol{x}_c^{(i)}\}$ for $i = 1, \ldots, N$. For simplicity, we denote the entire collection of $N$ pairs as $\{X_a, X_c\}$. Each audio segment and its corresponding text description are processed through separate encoders: $f_a(.)$ for audio and $f_c(.)$ for text. For a batch of size $N$, we have:

$$\boldsymbol{z}_a^{(i)} = f_a(\boldsymbol{x}_a^{(i)}) \in \mathbb{R}^d, \quad \boldsymbol{z}_c^{(i)} = f_c(\boldsymbol{x}_c^{(i)}) \in \mathbb{R}^d, \quad i = 1, \ldots, N$$

where $\boldsymbol{z}_a^{(i)}$ and $\boldsymbol{z}_c^{(i)}$ represent the audio and text encodings, respectively. To evaluate the similarity between embeddings $\boldsymbol{z}_a^{(i)}$ and $\boldsymbol{z}_c^{(i)}$, we calculate their similarity matrix as $C = \gamma \cdot (\boldsymbol{z}_c \boldsymbol{z}_a^\top)$. Here, $\tau$ is a scaling constant that adjusts the logarithmic scale after applying softmax, as detailed in part D. The similarity matrix $C$ is $\mathbb{R}^{N \times N}$, with $N$ compatible pairs along the diagonal and $N^2 - N$ non-compatible pairs elsewhere. The overall objective function is defined as $\mathcal{L} = 0.5 \cdot (\ell_{text}(C) + \ell_{audio}(C))$.where $\ell_{\text{text}}(C)$ and $\ell_{\text{audio}}(C)$ are computed separately for the text and audio embeddings, using cross-entropy loss. Specifically, $\ell_{\text{text}}(C) = -\frac{1}{N} \sum_{i=1}^{N} \log \frac{e^{(\boldsymbol{z}_c^{(i)} \cdot \boldsymbol{z}_a^{(i)}/\gamma)}}{\sum_{j=1}^{N} e^{(\boldsymbol{z}_c^{(i)} \cdot \boldsymbol{z}_a^{(j)}/\gamma)}}$. This promotes

joint optimization of the audio and text encoders along with their respective transformations, as described in later sections.

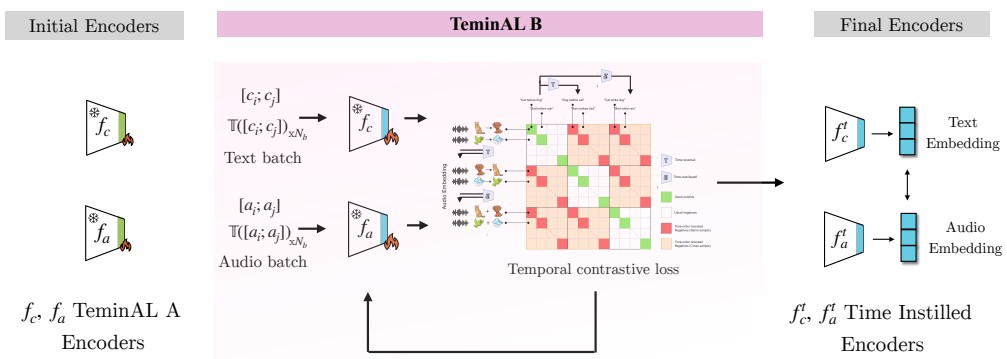

Figure 3: The overview of TeminAL B where we are post–training orginal CLAP encoders $f_c$ and $f_a$ with our **TeminAL** method to get $f_c^t$ and $f_a^t$. The functions as described in section 3.3, while the objective formulation for training $(f_c, f_a)$ to achieve $(f_c^t, f_a^t)$ has been described in section 3.4. The "Temporal contrastive loss" for TeminAL B has been elaborated in fig. 4.

.

### 3.4 OBJECTIVE FUNCTION FOR TEMINAL: WHAT ADDITION WE PROPOSE ON CLASSICAL CONTRASTIVE LEARNING

We propose a multi-stage training approach, outlined in fig. 1, with two stages: TeminAL A and TeminAL B. In TeminAL A, the model is trained to distinguish between single and multiple sounds, while in TeminAL B, the model learns to differentiate temporally distinct sounds along with their corresponding text labels. Both stages use contrastive learning with a modified infoNCE loss function (Oord et al., 2018), detailed further in this section and appendix B.3, the difference in training being the training data and contrastive objective. We have already elaborated on the different training dataset and it's formulation in section 3.2, while the detailed loss function for both stages has been discussed in this section. Context (text) and audio encodings are processed through their respective encoders, producing embeddings $C_e$ and $A_e$ as shown in fig. 3. These embeddings are used to form a (batch × batch) matrix to identify positive and negative pairs (see Figures 4). Similarity scores are calculated from these embeddings and used to compute the modified infoNCE loss function, as described latter in the section. Logits derived from similarity scores are transformed using a Softmax function to generate probabilities (equation 4 and equation 5), which are evaluated with cross-entropy against true labels. The loss function is computed as the sum of text loss ($L_c$) and audio loss ($L_a$), which sum up to form $L_B = L_{c_B} + \beta(L_{a_B})$ which stands for the TeminAL B loss. The text loss $L_{c_B}$ optimizes the selection of texts from $n$ possible options generated by $C_e$ (equation 3), while audio loss $L_{a_B}$ does the same for the audio embeddings (equation 7). This dual-component loss ensures balanced training of both context ($C_e$) and audio ($A_e$) encoders. The overall methodology is schematically depicted in Figure 3.

Unlike traditional contrastive loss functions that primarily reinforce true positives, our approach modifies the infoNCE loss to make encoders more sensitive to time-reversed and overlapping samples, as shown in equation 4 and equation 5. For temporal alignment, we use an adapted version of the InfoNCE loss function in both TeminAL A and B to distinguish the temporal sequence of audio-text pairs. For a time-aligned audio-text pair $(a, c)$, following section 3.1, we design a loss function that maintains chronological order within the pair, differing only in the loss components. The training batch for TeminAL A is defined as $B_{c_A} = \{B_{c_s}, B_{c_d}\}$ for texts (single and dual stitched audio) and $B_{a_A} = \{B_{a_s}, B_{a_d}\}$ for audio, following the conventions in section 3.2. For TeminAL B, the batches are $B_{c_B} = \{B_{c_f}, B_{c_r}, B_{c_o}\}$ for texts and $B_{a_B} = \{B_{a_f}, B_{a_r}, B_{a_o}\}$ for audio (forward, reversed, and time-overlaid); In general we represent batches of audio–text data by symbol $B$. These batches are processed through encoders, converting them into audio and text embeddings that are

used in subsequent stages of training. For the layout of the batches of data kindly refer to fig. 5. Furthermore, our encoders are not trained from scratch; we extend our framework using a pre-existing audio-language model comprising an audio encoder $f_{c_\theta}$ and a text encoder $f_{a_\phi}$ shown in fig. 3 from CLAP by Elizalde et al. (2023). These pre-trained encoders are post-trained to enhance temporal accuracy while maintaining baseline retrieval performance, as demonstrated in table 1. Due to limited dataset size, selective refinement of specific layers within $\Theta = \{\theta, \phi\}$ is performed, as schematically shown in fig. 1 and detailed in appendix B.3.

$$L_{c_B} = \sum_{(a,c) \in B_{c_B}} (\text{TNCE}(\boldsymbol{z}_a, \boldsymbol{z}_c) + \text{TNCE}(\boldsymbol{z}_a, \boldsymbol{z}_{\mathbb{T}(c)}) + \text{TNCE}(\boldsymbol{z}_a, \boldsymbol{z}_{\mathbb{O}(c)})) \tag{3}$$

To complete our model construct, in the following section we have explained the details of the loss function mathematically in equations equation 3–equation 9. Kindly note that, the hyperparameters introcuded are discussed in the following section 3.5. Earlier, we had seen the discussion on text and audio losses ($L_c$ and $L_a$), we now define them mathematically in the following equations. Here, TNCE stands for Temporal Noise Contrastive Estimation, a variant of the NCE loss.

$$\text{TNCE}(\boldsymbol{z}_a, \boldsymbol{z}_c) = -\log \frac{\exp(\boldsymbol{z}_a \cdot \boldsymbol{z}_c)}{\sum_{c' \in B_{c_f}} \exp(z_a \cdot \boldsymbol{z}_{c'}) + C^{c_r} + C^{c_o}} \tag{4}$$

$$\text{TNCE}(\boldsymbol{z}_a, \boldsymbol{z}_{\mathbb{O}(c)}) = -\log \frac{\exp(\boldsymbol{z}_a \cdot \boldsymbol{z}_{\mathbb{O}(c)})}{\sum_{c' \in B_{c_o}} \exp(\boldsymbol{z}_a \cdot \boldsymbol{z}_{\mathbb{O}(c')}) + C^{c_c}} \tag{5}$$

In equation 4, $C^{c_r}$ and $C^{c_o}$ is an accumulation of negatives fashioned via time-reversal and time–overlay respectively in equation 4, and is expressed as: $C^{c_r} = \alpha_{s_t} \exp(\boldsymbol{z}_a \cdot \boldsymbol{z}_{\mathbb{T}(c)}) + \alpha_{c_t} \sum_{c' \in B_{c_r} \setminus \{c\}} \exp(\boldsymbol{z}_a \cdot \boldsymbol{z}_{\mathbb{T}(c')})$ and $C^{c_o} = \alpha_{s_o}(\exp(\boldsymbol{z}_a \cdot \boldsymbol{z}_{\mathbb{O}(c)})) + \alpha_{t_o} \sum_{c' \in B_c \setminus \{o\}} \exp(\boldsymbol{z}_a \cdot \boldsymbol{z}_{\mathbb{O}(c')})$. While $C^{c_c}$ from equation 5 is expressed as:

$$C^{c_c} = \exp(\boldsymbol{z}_a \cdot \boldsymbol{z}_c) + \sum_{c' \in B_{c_f} \setminus \{c\}} \exp(\boldsymbol{z}_a \cdot \boldsymbol{z}_{c'}) + \alpha_{s_t} \exp(\boldsymbol{z}_a \cdot \boldsymbol{z}_{\mathbb{T}(c)}) + \alpha_{c_t} \sum_{c' \in B_{c_r} \setminus \{c\}} \exp(\boldsymbol{z}_a \cdot \boldsymbol{z}_{\mathbb{T}(c')}) \tag{6}$$

The loss function is constructed in such a way that it penalises the miss-classifications among the audio–text pairs. The loss formulations gives a handle on penalising the time-reversed samples and time–overlayed samples with the hyper–parameters $\alpha_{s_t}$ and $\alpha_{s_o}$, we present a detailed analysis on effects of these hyper–parameters later in section 3.5. The total loss $L_B$ for TeminAL B can then be written with $L_{c_B}$ and $L_{a_B}$ which follows the same formulation as $L_{c_B}$. Detailed formulation for $L_{c_B}$ and $L_{a_B}$ have been provided in the supplementary section. The net loss for TeminAL B is expressed as $L_B = L_{c_B} + \beta(L_{a_B})$, where $L_{a_B}$ is as follows:

$$L_{a_B} = \sum_{(a,c) \in B_{a_B}} (\text{TNCE}(z_c, z_a) + \text{TNCE}(z_{\mathbb{T}(c)}, z_a) + \text{TNCE}(z_{\mathbb{O}(c)}, z_a)) \tag{7}$$

After discussing the loss formulation of TeminAL B, we have similar formulation for TeminAL A. With necessary changes in the configuration of data within the batch ($B_{a_A}$ and $B_{c_A}$) as it's mentioned in the previous paragraph, kindly refer to fig. 6 for the layout of the batch. The mathematical formulation of the contrastive loss function is described as follows and schematically shown in fig. 7:

$$L_{c_A} = \sum_{(\mathbb{T}(a), \mathbb{T}(c)) \in B_{c_A}} (\text{TNCE}(\boldsymbol{z}_{c_s}, \boldsymbol{z}_{a_s}) + \text{TNCE}(\boldsymbol{z}_{c_d}, \boldsymbol{z}_{a_d})) \tag{8}$$

$$L_{a_A} = \sum_{(\mathbb{O}(a), \mathbb{O}(c)) \in B_{a_A}} (\text{TNCE}(\boldsymbol{z}_{a_s}, \boldsymbol{z}_{c_s}) + \text{TNCE}(\boldsymbol{z}_{a_d}, \boldsymbol{z}_{c_d})) \tag{9}$$

The loss function construction and mathematical derivation of $L_A = L_{c_A} + \beta_A(L_{a_A})$ for TeminAL A is also detailed in appendix B.3.

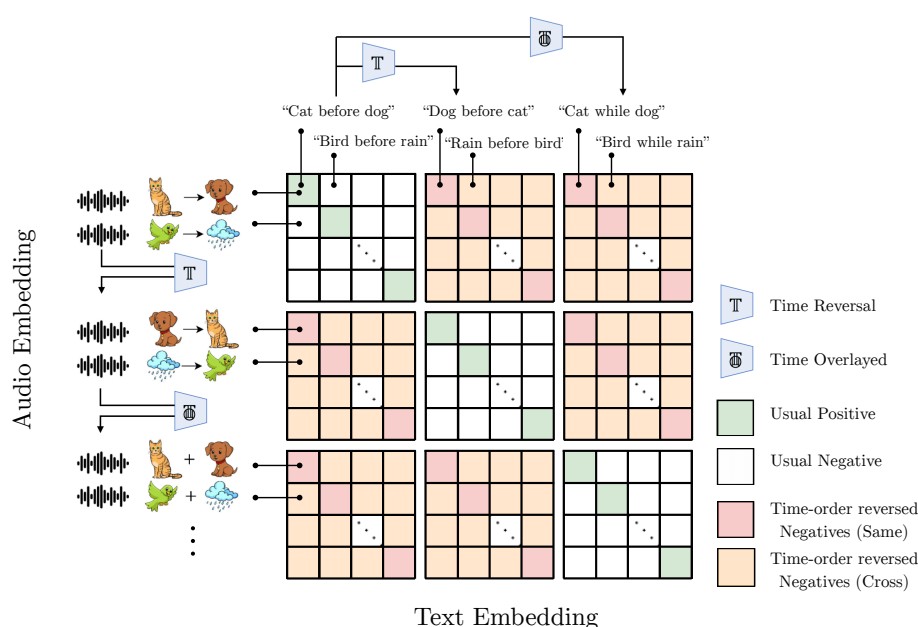

Figure 4: The schematic showing Temporal Contrastive Loss for TeminAL B. On the vertical axis we have the audio embeddings with batches of data corresponding to $B_{a_B} = \{B_{a_f}, B_{a_r}, B_{a_o}\}$ and text embedding batches of data corresponding to $B_{c_B} = \{B_{c_f}, B_{c_r}, B_{c_o}\}$ on the horizontal axis.

### 3.5 DETAILS ON HYPER-PARAMETERS OF THE LOSS FORMULATION

The loss function formulation introduces hyper-parameters that affect the temporal sensitivity of the encoders. The choice of $\{\alpha_{s_o}, \alpha_{c_o}, \alpha_{s_t}, \alpha_{c_t}\}$ and $\{\beta_A, \beta\}$, which can be either 0 or 1, significantly impacts the model's behavior. For example, setting $\alpha_{s_t} = 1$ increases sensitivity to time-reversed sound samples by adding the term $\alpha_{s_t} \exp(z_a \cdot z_{\mathbb{T}(c)})$ to the expression $C_{tr}$ denominator of equation 6. This adjustment forces the encoders to ensure the sum of terms equals unity, which reduces the encoding values of non-similar pairs, enhancing sensitivity to time-reversed samples and guiding the encoders to assign lower values to dissimilar batch samples.

These coefficients also help adapt the model to different datasets. In fig. 4, these parameters extend the contrastive loss function over time: the top three sub-squares represent TNCE($z_a, z_c$), the middle sub-squares represent TNCE($z_a, z_{\mathbb{T}(c)}$), and the last three sub-squares represent TNCE($z_a, z_{\mathbb{O}(c)}$). The top-left quadrant shows contrastive loss on stitched pairs, with positive (green) and negative (red) diagonal terms crucial for temporal understanding.

The key role of $\beta$ is to increase the number of training samples, while the $\alpha$ coefficients enhance sensitivity to time-reversal and overlapping sounds. When $\alpha = 0$, sensitivity is nullified, but higher values compel the encoders to refine recognition of temporal variations by minimizing the denominator. Without these hyper-parameters, the loss converges similarly, but encoders learn different relationships. Their inclusion ensures distinct sample treatment, enhancing temporal sensitivity.

## 4 EXPERIMENTS

### 4.1 BASE MODEL

We employ a pre–trained CLAP model Elizalde et al. (2023) using transformer-based encoders: HTSAT Chen et al. (2022) for audio and BERT Devlin et al. (2018) for text, each with a projection layer. We focuses on the final layers and projection layers of both encoders for our training. While our TeminAL training approach is model-agnostic, we start with the CLAP model as a foundation.

## 4.2 ZSTE AND DOWNSTREAM TASKS

We construct comprehensive evaluation of our proposed method in order to satisfy our objectives of temporal instillation. ZSTE (**Z**ero **S**hot **T**emporal **E**valuation) is our evaluation framework for assessing contrastive models in zero-shot tasks. The implementation is discussed in algorithm 2 and algorithm 3. ZSTE begins with basic classification on unseen classes in Task 1 (fig. 10), progressing to complex scenarios involving overlapping audio features and novel composite text classes in Task 2. The subtasks 2A and 2B involve the model being able to distinguish both classes and at–least one class respectively, model which correctly understands both sounds performs well on subtask A. The subtask 2C and 2D are similar tasks as 2A and 2B but on overlapping sounds rather than concatenated sounds. Thus given the overlapping text-audio pair, we need compute the accuracy of the model to detect both the audio classes in 2C and at-least one of the classes in 2D (fig. 11). Models which are able to distinguish multiple overlapping sounds, perform better in 2C.

ZSTE Task 3 evaluates temporal comprehension by testing sequences of interchanged acoustic events, building on the configuration from Task 2 but now using temporal texts (fig. 12). A model capable of understanding temporal relationships in text will excel in this task. Task 4 (fig. 13) assesses the model's ability to maintain focus amid irrelevant class labels, which act as noise to the actual audio embeddings. Task 5 (fig. 14) examines the model's generalization to out-of-distribution prompts, reflecting real-world complexities. Each of these three tasks includes subtasks A and B: Subtask A requires detecting all audio classes, while Subtask B involves identifying at least one class. These tasks test the model's grasp of temporality and general language attribution. Models with a robust understanding of both temporality and language generalisability will perform better.

This comprehensive approach ensures robust evaluation of the model's zero-shot learning capabilities. The primary aim is to foster model improvement rather than solely benchmark performance. Further details on ZSTE is shown in appendix B.4.

## 5 RESULTS

In this section, we present the experimental results to support the claims outlined in our objectives and discuss our key findings. The results are organized around several downstream tasks, beginning with audio and text retrieval and progressing to an in-depth evaluation of the models' temporal behavior and finally comparing various SOTA contrastive ALM models for temporal understanding. Firstly we compare the retrieval performance of closed-ended and open-ended models on benchmark AudioCap and Clotho dataset, as summarized in Tables 1 and 7. Our model, T–CLAP, demonstrates superior performance across retrieval tasks, surpassing most existing models in both closed-ended and open-ended categories. Notably, it achieves competetive state-of-the-art results for both text-to-audio (T–A) and audio-to-text (A–T) retrieval tasks. These results underscore the effectiveness of our contrastive training strategy, employed both during the pre-training phase of CLAP and the subsequent fine-tuning phase using TeminAL. Our approach effectively preserves the contrastive knowledge acquired during pre-training, ensuring strong retrieval performance. However, as mentioned previously in section 1, the retrieval metrics alone do not fully encapsulate the temporal understanding capabilities of our model. To address this limitation, we conduct a rigorous Zero-Shot Temporal Evaluation (ZSTE), which offers deeper insights into the temporal reasoning ability of T–CLAP. The method of evaluation is further elaborated in appendix B.4.

Firstly we try and study the model's behaviour through the hyperparameter variations, the role of each hypermeter is detailed in section 3.5. The results from Table 2 convey important information on how the model captures temporal behaviour in general (across the ZSTE tasks) through our modification of the overall objective function through parametric variations and the impact of including a multistage training objective. As mentioned in section 4.2 in Task 1, the model must excel in the initial pre-training task, and results indicate that our training strategies prevent catastrophic forgetting, although increasing temporality in the objective function we observe decrease in performance in this task we still remain well above across different tasks. Simillarly, task 2 tests multi-sound understanding, where the two-stage TeminAL AB training significantly improves sound distinction capabilities. Task 3 focuses on temporal reasoning, demonstrating that specific loss coefficients enhance the model's ability to capture temporal relationships. Task 4 and 5 evaluates complex and general text prompts, showing that our model outperforms the original CLAP in correctly mapping stitched audio to appropriate text although much room for improvement is there for future models.

| Model | T-A Retrieval | | | A-T Retrieval | | |
|---|---|---|---|---|---|---|
| | R@1 | R@5 | R@10 | R@1 | R@5 | R@10 |
| MMT | 36.1 / 6.7 | 72.0 / 21.6 | 84.5 / 33.2 | 39.6 / 7.0 | 76.8 / 22.7 | 86.7 / 34.6 |
| ML-ACT | 33.9 / 14.4 | 69.7 / 36.6 | 82.6 / 49.9 | 39.4 / 16.2 | 72.0 / 37.6 | 83.9 / 50.2 |
| CLAP | 34.6 / 16.7 | 70.2 / 41.1 | 82.0 / 54.1 | 41.9 / 20.0 | 73.1 / 44.9 | 84.6 / 58.7 |
| CLAP-LAION | **36.2 / 17.2** | 70.3 / 42.9 | 82.5 / 55.4 | 45.0 / **24.2** | 76.7 / 51.1 | 88.0 / 66.9 |
| CompA–CLAP | 36.1 / 16.8 | **78.6 / 43.5** | **90.2 / 56.1** | 47.8 / 23.9 | 83.5 / 50.7 | **90.2 / 67.6** |
| **T–CLAP(ours)** | 35.1 / 17.0 | 71.2 / 42.2 | 82.1 / 54.7 | **49.2** / 23.1 | **85.1 / 52.2** | 87.8 / 66.4 |

Table 1: Comparison of models on text-to-audio (T-A) and audio-to-text (A-T) retrieval tasks. Performance of Text-to-Audio and Audio-to-Text retrieval on AudioCap/Clotho dataset. The values for other models have been taken from previous publications (Ghosh et al., 2023; Elizalde et al., 2023).

| | Loss–coefficients | | | | | ZSTE | | | | | | | | | |
|---|---|---|---|---|---|---|---|---|---|---|---|---|---|---|---|
| | | | | | | Task 1 | Task 2 | | | | Task 3 | | Task 4 | | Task 5 | |
| | $\alpha_{s_t}$ | $\alpha_{c_t}$ | $\alpha_{s_o}$ | $\alpha_{c_o}$ | $\beta$ | A | A | B | C | D | A | B | A | B | A | B |
| T–B | 0 | 0 | 0 | 0 | 1 | 78.77 | 10.11 | 77.60 | 8.46 | 78.01 | 32.67 | 31.57 | 3.50 | 1.12 | 26.01 | 1.10 |
| | 1 | 0 | 1 | 0 | 1 | 77.67 | 11.02 | 83.20 | 8.71 | 81.21 | 48.50 | 18.20 | 36.28 | 7.10 | 27.07 | 12.7 |
| | 0 | 1 | 0 | 1 | 1 | 76.54 | 10.11 | 83.44 | 7.98 | 83.01 | 49.11 | 18.74 | 40.38 | 8.32 | 28.01 | 15.2 |
| | 1 | 1 | 1 | 1 | 1 | 76.14 | 12.20 | 83.61 | 11.44 | 83.2 | 51.3 | 22.83 | 41.10 | 9.18 | 31.21 | 15.8 |
| T–AB | 0 | 0 | 0 | 0 | 1 | 77.34 | 38.45 | 80.90 | 49.87 | 79.67 | 34.22 | 33.12 | 34.50 | 32.15 | 27.34 | 2.01 |
| | 1 | 0 | 1 | 0 | 1 | 76.76 | 39.23 | **86.34** | 50.65 | 83.78 | 52.12 | 42.45 | 39.45 | 38.67 | 28.45 | 14.23 |
| | 0 | 1 | 0 | 1 | 1 | 75.29 | 43.45 | 85.89 | 59.56 | 84.56 | 50.78 | 24.33 | 52.78 | 41.45 | 29.78 | 16.89 |
| | 1 | 1 | 1 | 1 | 1 | 75.11 | **46.78** | 86.12 | **62.34** | 85.45 | **54.56** | **56.78** | **46.23** | **44.89** | **32.45** | **18.34** |

Table 2: Hyper-parameter analysis for loss coefficients $\{\alpha, \beta\} = \{\alpha_{s_t}, \alpha_{c_t}, \alpha_{s_o}, \alpha_{c_o}, \beta\}$. Each Task is defined according to appendix B.4, kindly refer this section for details on each task. Here **T–B** and **T–AB** refers to models trained with only TeminAL B and TeminAL A + B respectively.

Our findings show that setting $\alpha_{c_t} = 0$ benefits ZS-tasks with fewer confusing classes, like Task 1, while $\alpha_{c_t} = 1$ improves performance in tasks with more complex classes (Tasks 4, and 5). Models trained with $\alpha_{s_o} = 0$ lack overlaid classes in the denominator, impacting how negatives are penalized. Table 2 illustrates our model's adaptability across ZSTE tasks, with key improvements noted when using combined TeminAL A and B training. Adjusting $\alpha_{c_t}$ and $\alpha_{c_o}$ makes the model more sensitive to time-reversed samples, enhancing performance in time-sensitive tasks. The T–CLAP model sometimes struggles with sound distinction due to training focused on temporal understanding, not sound separation, affecting sensitivity and overall accuracy. However, hierarchical training with both TeminAL A and B significantly improves sound distinction and general language understanding tasks. This result grounds the importance of our multi-stage training method in order to learn the temporal behavior of multiple sound as described in section 1.

| Tasks | Subtasks | ML-ACT | CLAP | CLAP-LAION | CompA-CLAP | T-CLAP | |
|---|---|---|---|---|---|---|---|
| | | | | | | TeminAL B | TeminaAL AB |
| **1** | **A** | 76.12 | 81.22 | 82.5 | **83.0** | 76.14 | 75.11 |
| **2** | **A** | 7.2 | 9.59 | 10.1 | 18.4 | 32.20 | **46.78** |
| **2** | **B** | 78.1 | 81.00 | 81.3 | **91.6** | 83.61 | 87.12 |
| **2** | **C** | 6.5 | 9.39 | 10.0 | 21.3 | 31.4 | **62.34** |
| **2** | **D** | 71.7 | 80 | 80.4 | **90.8** | 83.2 | 85.45 |
| **3** | **A** | 28.01 | 33.27 | 34.93 | **54.5** | 51.3 | **54.56** |
| **3** | **B** | 27.5 | 34.29 | 34.6 | 49.87 | 22.83 | **56.78** |
| **4** | **A** | 2.2 | 2.4 | 7.56 | **48.71** | 41.1 | 46.23 |
| **4** | **B** | 2.0 | 1.98 | 5.45 | 38.74 | 9.18 | **44.89** |
| **5** | **A** | 3.0 | 26 | 26.4 | **36.81** | 31 | 32.45 |
| **5** | **B** | 2.5 | 0 | 0.7 | 18.2 | 15.8 | **18.34** |

Table 3: Showing the comparison of various contrastive learning models on our ZSTE tasks. The details on each task is discussed in section 4.2 and further detailed in appendix B.4.

We next evaluate and compare the performance of various models on temporal understanding through the ZSTE tasks. As shown in Table 3, T–CLAP outperforms most state-of-the-art models across a majority of tasks. Notably, T–CLAP excels in tasks 2A and 2C, as mentioned previously section 4.2, these two substasks invovle the model to distinguish multiple sounds and detecting both the sounds. While remaining competitive in Task 1A we observe, demonstrating that our temporal instillation approach effectively instills the model with a sense of time without significantly degrading its performance on the original pre-training tasks. Furthermore, results of Tasks 2B and 2C, which require the model to associate multiple sounds with at least one correct class, show better performance with larger models due to their capacity to handle complex associations. Following the results of Task 3, we observe T–CLAP performs competitively with other models, despite those models being specifically trained on audio-text pairs. Interestingly, these competing models achieve strong results on Task 3 without performing as well in tasks requiring the differentiation of multiple audio sounds, such as Tasks 2B and 2C. However, all models, including T–CLAP, encounter challenges in general language understanding tasks, such as Tasks 4 and 5. This suggests that leveraging a more robust pre-trained language encoder along with diversifying the dataset could further enhance overall performance.

## 6 Conclusion

This research introduces the Temporal Instillation in Audio-Language Models, a post-training technique that enhances temporal and language understanding in Audio-Language Models (ALMs). Our approach employs sequential inversion and temporal augmentations, effectively improving sequential discernment in ALMs. The hierarchical training strategy proves crucial, as seen in the performance comparison between TeminAL B and TeminAL AB, highlighting the need for structured training in complex tasks like time instillation. Our findings also demonstrate that modifying the infoNCE loss improves model sensitivity, as shown in our parametric study (table 2). Zero Shot Temporal Evaluation (ZSTE) results (section 5) confirm T–CLAP's strength in zero-shot classification and retrieval tasks, offering new evaluation insights for contrastive learning models. While T–CLAP shows a slight decrease in traditional audio classification accuracy, it consistently outperforms baseline models in scenarios involving temporal relationships, demonstrating enhanced sequential information processing. This study opens new research directions, particularly in refining contrastive loss for broader task optimization, paving the way for ALMs that excel in both retrieval and complex temporal-linguistic tasks.

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

## A  APPENDIX

## B  SUPPLEMENTARY SECTION

### B.1  PROOF OF PROPOSITIONS

#### B.1.1  PROPOSITION 1 :

Contrastive models, when used for audio-text matching, do not comprehend the semantic relationship between the audio and text, but rather operate by matching similar audios to similar texts based on superficial features.

Let $f_{audio} : A \to \mathbb{R}^d$ and $f_{text} : T \to \mathbb{R}^d$ be the functions mapping audio $A$ and text $T$ into a d-dimensional embedding space, respectively. The similarity score between an audio sample $a$ and a text sample $t$ is given by $s(a, t) = f_{audio}(a) \cdot f_{text}(t)$.

1. Contrastive models can yield high similarity scores for pairs of audio and text samples that share similar superficial features but lack semantic congruence.

2. Contrastive models, as defined, cannot inherently discern semantic relationships between audio and text but rely on the co-occurrence of similar features in their respective embeddings.

Assume a pair of audio samples $a_1, a_2$ and text samples $t_1, t_2$ such that $a_1$ and $t_1$, $a_2$ and $t_2$ are semantically congruent but share similar superficial features with $a_2$ and $t_1$ respectively.

According to the model, $s(a_1, t_1)$ and $s(a_2, t_2)$ should be high. However, due to the shared superficial features, $s(a_1, t_2)$ and $s(a_2, t_1)$ may also be high, indicating a false positive match.

This contradiction shows that the model's high similarity score does not necessarily correspond to a true semantic match, supporting the hypothesis.

### B.2  DATASET SELECTION AND CREATION

For dataset selection and creation process we chose ESC–50 dataset. Due to its high audio quality, adequate pre–processing, suitable length and number of samples, and its inherent robustness. Importantly, we excluded datasets generated through crowd–sourcing to reduce labeling inaccuracies. ESC–50's assortment of 50 classes encompasses a variety of real-world sounds from natural, animal, and human sources, making it versatile for different applications and particularly effective for zero-shot classification tasks, which require identifying items from previously unseen categories. From the ESC–50 dataset we get 50 pairs of Audio, Label data, these pairs are then processed according to algorithm 2 to make a training dataset. We select two distinct sounds from the possible 50 sounds giving us a total of 2450 pairs $(a_i, a_j)$ and $(t_i, t_j)$ of sounds. For each pair we have 3 possible configurations using keywords 'before' , 'after' and 'while' as suggested in section 3.1. Thus our total dataset thus becomes 7350 data pairs. For teminAL A, we only use 2450 pairs of data while selecting either one of the audio and text from this pair. The prompt used for concatenating the texts are 'single sound of $t_i$' and 'combined sound of $t_i$ and $t_j$'.

Our Sequential Inversion Approach challenges traditional contrastive learning methods, which typically align audio segments with matching text while contrasting them against unrelated pairs. This practice, akin to a bag-of-words model, often fails to capture sequential nuances as it emphasizes distinguishing features over temporal understanding. To foster a deeper comprehension of sequences, we introduced a novel technique for generating negative samples that share thematic elements, compelling the model to focus on the order of events. This method, depicted in fig. 2, utilizes two types of temporal augmentations "before" and "while" to enhance the model's ability to discern sequential information. The transformation aims to capture the dynamic interplay between the two arguments, allowing the model to discern the original audio-transcript pair $(a, c)$ from its transformed versions $(a, \mathbb{O}(c))$ and $(\mathbb{O}(a), c)$. It is applicable to concatenated audio or transcription pairs, effectuating a temporal reordering of the components.

---

**Algorithm 1** Dataset Preparation and Sequential Inversion for Contrastive Learning

---

1: **Input:** Acoustic Events Dataset $P = \{(\text{Audio}_i, \text{Label}_i)\}$
2: **Output:** Refined Dataset $(A_{pos}, A_{neg}, T_{pos}, T_{neg})$ for Contrastive Learning
3: **Initialization:**
4: Initialize ESC-50 dataset with selected criteria.
5: Organize dataset into classes $\{C_1, C_2, \ldots, C_n\}$ representing distinct sounds.
6: Define empty sets for positive and negative samples: $A_{pos}, A_{neg}, T_{pos}, T_{neg}$.

7: **Step 1: Positive Sample Selection**
8: **for** each class $C_i$ in the dataset **do**
9:     **for** each audio-text pair $(a_j, c_j) \in C_i$ **do**
10:         **if** $(a_j, c_j)$ meets quality standards **then**
11:             Append $a_j$ to $A_{pos}$, $c_j$ to $T_{pos}$.
12:         **end if**
13:     **end for**
14: **end for**

15: **Step 2: Sequential Inversion and Overlay**
16: **for** each $(a_j, c_j) \in (A_{pos}, T_{pos})$ **do**
17:     Generate negative audio samples $a'_j$ using inversion function $\mathbb{T}$.
18:     Define $\mathbb{T}(a) = [a_j \oplus a_i]$, $\mathbb{T}(c) = [c_j; \tau_t; c_i]$.
19:     Generate overlapping samples using overlay function $\mathbb{O}$:
        $\mathbb{O}(a) = [a_j \wedge a_i]$, $\mathbb{O}(c) = [c_j; \tau_o; c_i]$.
20:     Append resulting negative samples $a'_j, c'_j$ to $A_{neg}$ and $T_{neg}$.
21: **end for**

22: **Step 3: Template-Based Caption Generation**
23: **for** each positive sample $c_k \in T_{pos}$ **do**
24:     Generate captions using $\text{Caption}(c_k)$ and append to $T_{pos}$.
25: **end for**
26: **for** each negative sample $c_n \in T_{neg}$ **do**
27:     Generate captions using $\text{Caption}(c_n)$ and append to $T_{neg}$.
28: **end for**

29: **Return:** $(A_{pos}, A_{neg}, T_{pos}, T_{neg})$.

---

## B.3 MATHEMATICAL DERIVATIONS:

In this section, we derive the loss functions used in our model, specifically focusing on the Temporal Noise Contrastive Estimation (TNCE) technique. TNCE is a variant of the Noise Contrastive Estimation (NCE) loss, adapted for temporal learning tasks. This method helps in effectively distinguishing between positive and negative samples over time. (Kindly note that we have used '$t$' as text in the Mathematical derivation instead of '$c$' as we have shown in the main paper, all the other component remains the same. For example here we have shown batch of texts $B_t = \{B_{t_f}, B_{t_r}, B_{t_o}\}$ instead of $B_c = \{B_{c_f}, B_{c_r}, B_{c_o}\}$).

For the loss function $L_r$, we define it as follows:

$$L_{t_B} = \sum_{(a,t) \in B} (\text{TNCE}(\boldsymbol{z}_a, \boldsymbol{z}_t) + \text{TNCE}(\boldsymbol{z}_{\mathbb{T}(t)}, \boldsymbol{z}_a)) + \text{TNCE}(\boldsymbol{z}_{\mathbb{O}(t)}, \boldsymbol{z}_a)) \tag{10}$$

Here, $\text{TNCE}(\boldsymbol{z}_a, \boldsymbol{z}_t)$, $\text{TNCE}(\boldsymbol{z}_{\mathbb{T}(t)}, \boldsymbol{z}_a))$ and $\text{TNCE}(\boldsymbol{z}_{\mathbb{O}(t)}, \boldsymbol{z}_a))$ represent the temporal consistent, temporally reversed and temporally overlap components of the TNCE loss, respectively. The function TNCE is calculated by the formula:

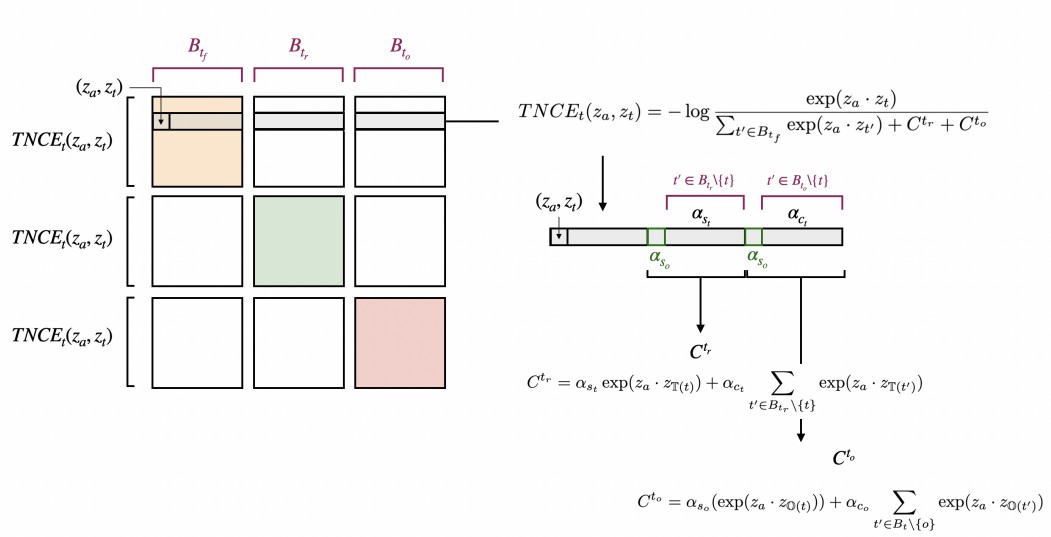

Figure 5: Schematic explanation of the terms in loss function for TeminAL B. Here we show a term (row) in the summation of $L_{t_B}$ which is $\text{TNCE}_t(\boldsymbol{z}_a, \boldsymbol{z}_t)$ The other two terms $\text{TNCE}_t(\boldsymbol{z}_a, \boldsymbol{z}_t)$ and $\text{TNCE}_t(\boldsymbol{z}_a, \boldsymbol{z}_t)$ of this loss function can be calculated in the similar way and will belong to the green and pink blocks of the above schematic. Here, $B_{t_f}, B_{t_r}$ and $B_{t_o}$ are the batches of texts corresponding to time consistent, reversed and overlaid samples which compose the whole batch of text following the same convention as shown in section 3.4.

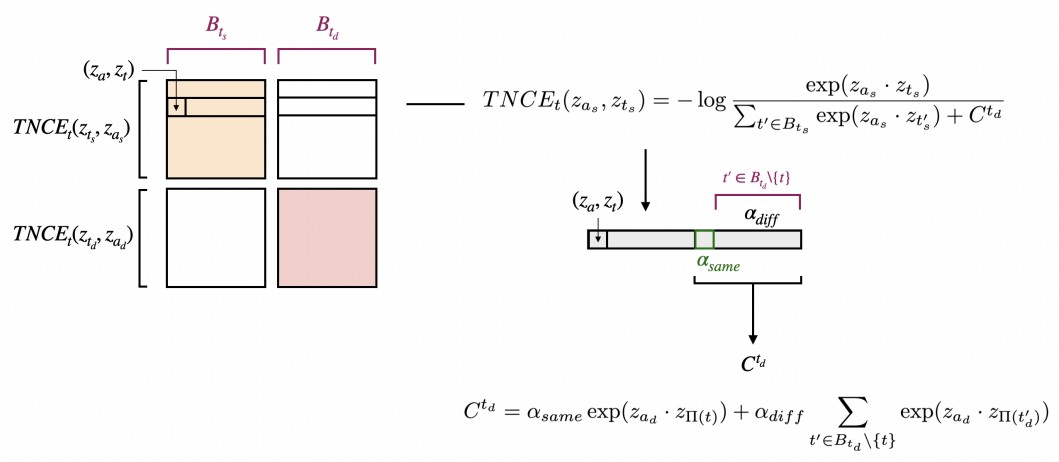

Figure 6: Schematic explanation of the terms in loss function for TeminAL A. Here we show a term (row) in the summation of $L_{t_A}$ which is $\text{TNCE}_t(\boldsymbol{z}_{a_s}, \boldsymbol{z}_{t_s})$ The other term $\text{TNCE}_t(\boldsymbol{z}_{a_d}, \boldsymbol{z}_{t_d})$ of this loss function can be calculated in the similar way and will belong to the green block of the above schematic. Here, $B_{t_s}$ and $B_{t_d}$ are the batches of texts corresponding to single and concatenated (double) samples which compose the whole batch of text following the same convention as shown in section 3.4.

$$\text{TNCE}(\boldsymbol{z}_a, \boldsymbol{z}_t) := -\log \frac{\exp(\boldsymbol{z}_a \cdot \boldsymbol{z}_t)}{\sum_{t' \in B_{t_f}} \exp(\boldsymbol{z}_a \cdot \boldsymbol{z}_{t'}) + C^{t_r} + C^{t_o}} \tag{11}$$

Similarly, the overlap component $TNCE_o$ is given by:

$$\text{TNCE}(\boldsymbol{z}_a, \boldsymbol{z}_t) := -\log \frac{\exp(\boldsymbol{z}_a \cdot \boldsymbol{z}_t)}{\sum_{t' \in B_{t_o}} \exp(\boldsymbol{z}_a \cdot \boldsymbol{z}_{t'}) + C^{t_c}} \tag{12}$$

In these equations: $B$ represents the batch of user-item pairs $(a, t)$, where $a$ is a user and $t$ is a temporal context. $\boldsymbol{z}_a$ and $\boldsymbol{z}_t$ denote the latent representations of the user and the temporal context, respectively. $B_{t_f}$ and $B_{t_o}$ are subsets of the batch $B$ that serve as temporal and overlap negatives, respectively. The constants $C^{t_r}$, $C^{t_o}$, and $C^{t_c}$ are designed to account for additional temporal and contextual information, enhancing the robustness of the loss function against trivial solutions. The term $C^{t_r}$ accounts for the influence of time-reversed negatives and is defined as:

$$C^{t_r} = \alpha_{s_t} \exp(\boldsymbol{z}_u \cdot z_{\Pi(t)}) + \alpha_{c_t} \sum_{t' \in B_{t_r} \setminus \{t\}} \exp(\boldsymbol{z}_u \cdot \boldsymbol{z}_{\Pi(t')}) \tag{13}$$

where: $\Pi(t)$ denotes the time-reversed representation of the context $t$. The coefficients $\alpha_{s_t}$ and $\alpha_{c_t}$ modulate the contribution of individual and cumulative time-reversed negatives, respectively. The term $C^{t_o}$ captures the effect of overlapping contexts, defined as:

$$C^{t_o} = \alpha_{s_o}(\exp(\boldsymbol{z}_a \cdot \boldsymbol{z}_t) + \exp(\boldsymbol{z}_a \cdot \boldsymbol{z}_{\Pi(t)})) + \alpha_{c_o} \sum_{t' \in B_t \setminus \{o\}} \exp(\boldsymbol{z}_a \cdot \boldsymbol{z}_{\Pi(t')}) \tag{14}$$

Here: $\alpha_{s_o}$ and $\alpha_{c_o}$ control the impact of single and multiple overlapping contexts.

Finally, $C^{t_c}$ integrates both temporal and contextual negative sampling:

$$C^{t_c} = \left( \exp(z_a \cdot \boldsymbol{z}_t) + \sum_{t' \in B_{t_f} \setminus \{t\}} \exp(z_a \cdot \boldsymbol{z}_{t'}) \right)$$
$$+ \left( \alpha_s \exp(\boldsymbol{z}_a \cdot \boldsymbol{z}_{\Pi(t)}) + \alpha_c \sum_{t' \in B_{t_r} \setminus \{t\}} \exp(\boldsymbol{z}_a \cdot \boldsymbol{z}_{\Pi(t')}) \right) \tag{15}$$

This term combines the effect of immediate and cumulative context influences, with parameters $\alpha_s$ and $\alpha_c$ providing tunable weights.

For the loss function $L_{a_B}$, which deals with another set of temporal dynamics, we follow a similar structure. The formulation and constants remain analogous, ensuring consistency across different temporal modeling aspects.

$$L_{a_B} = \sum_{(a,t) \in B} \left( \text{TNCE}(\boldsymbol{z}_a, \boldsymbol{z}_t) + \text{TNCE}(\boldsymbol{z}_{\mathbb{T}(t)}, \boldsymbol{z}_a) \right) + \text{TNCE}(\boldsymbol{z}_{\mathbb{O}(t)}, \boldsymbol{z}_a)) \tag{16}$$

Here, TNCE stands for Temporal Noise Contrastive Estimation, a variant of the NCE loss tailored for temporal learning, and is calculated as:

$$\text{TNCE}(\boldsymbol{z}_a, \boldsymbol{z}_t) = -\log \frac{\exp(\boldsymbol{z}_a \cdot \boldsymbol{z}_t)}{\sum_{t' \in B_{t_f}} \exp(\boldsymbol{z}_a \cdot \boldsymbol{z}_{t'}) + C^{t_r} + C^{t_o}} \tag{17}$$

$$\text{TNCE}(\boldsymbol{z}_{\mathbb{O}(t)}, \boldsymbol{z}_a)) = -\log \frac{\exp(\boldsymbol{z}_a \cdot \boldsymbol{z}_t)}{\sum_{t' \in B_{t_o}} \exp(\boldsymbol{z}_a \cdot \boldsymbol{z}_{t'}) + C^{t_c}} \tag{18}$$

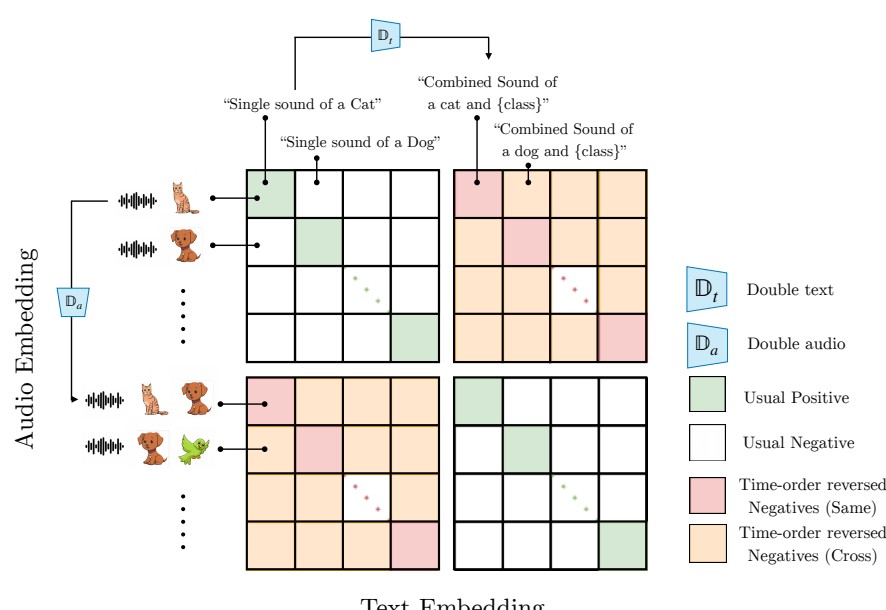

Figure 7: The schematic showing Temporal Contrastive Loss for TeminAL A. On the vertical axis we have the audio embeddings with batches of data corresponding to $B_{a_A} = \{B_{a_s}, B_{a_d}\}$ and text embedding batches of data corresponding to $B_{c_A} = \{B_{c_s}, B_{c_d}\}$ on the horizontal axis. Here, $\{B_{a_s}, B_{a_d}\}$ corresponds to batch of single audio and double audio respectively and similarly $\{B_{c_s}, B_{c_d}\}$ corresponds to batch of single text and double text respectively.

In this expression, $B$ represents the batch of $(a, t)$ pairs, and $B_t$ is the set of text samples within the batch that serve as temporal negatives. $C^t$ is an accumulation of negatives fashioned via time-reversal, and is expressed as:

$$C^{t_r} = \alpha_{s_t} \exp(\boldsymbol{z}_a \cdot \boldsymbol{z}_{\Pi(t)}) + \alpha_{c_t} \sum_{t' \in B_{t_r} \backslash \{t\}} \exp(\boldsymbol{z}_a \cdot \boldsymbol{z}_{\Pi(t')}) \tag{19}$$

$$C^{t_o} = \alpha_{s_o}(\exp(\boldsymbol{z}_a \cdot \boldsymbol{z}_t) + \exp(\boldsymbol{z}_a \cdot \boldsymbol{z}_{\Pi(t)})) + \alpha_{c_o} \sum_{t' \in B_t \backslash \{o\}} \exp(\boldsymbol{z}_a \cdot \boldsymbol{z}_{\Pi(t')}) \tag{20}$$

$$C^{t_c} = \left( \exp(\boldsymbol{z}_a \cdot \boldsymbol{z}_t) + \sum_{t' \in B_{t_f} \backslash \{t\}} \exp(\boldsymbol{z}_a \cdot \boldsymbol{z}_{t'}) \right) +$$
$$\left( \alpha_s \exp(\boldsymbol{z}_a \cdot \boldsymbol{z}_{\Pi(t)}) + \alpha_c \sum_{t' \in B_{t_r} \backslash \{t\}} \exp(\boldsymbol{z}_a \cdot z_{\Pi(t')}) \right) \tag{21}$$

Now we move on towards deriving the mathematical formulations for TeminAL A. Following from our initial discussion from section 3.4. For the loss function $L_{t_A}$, we define it as follows:

$$L_{t_A} = \sum_{(\mathbb{T}(u), \mathbb{T}(t)) \in B} (\text{TNCE}(\boldsymbol{z}_{t_s}, \boldsymbol{z}_{a_s}) + \text{TNCE}(\boldsymbol{z}_{t_d}, \boldsymbol{z}_{a_d})) \tag{22}$$

Here, $\text{TNCE}(\boldsymbol{z}_{t_s}, \boldsymbol{z}_{a_s})$ and $\text{TNCE}(\boldsymbol{z}_{t_d}, \boldsymbol{z}_{a_d})$ represent the temporal and overlap components of the TNCE loss, respectively. $\boldsymbol{z}_{t_s}, \boldsymbol{z}_{a_s}$ represents the text and audio samples of single samples in the

batch. And $\boldsymbol{z}_{t_d}, \boldsymbol{z}_{a_d}$ represents the text and audio samples of the double or concatenated batch. The function TNCE is calculated by the formula:

$$\text{TNCE}(\boldsymbol{z}_{a_s}, \boldsymbol{z}_{t_s}) = -\log \frac{\exp(\boldsymbol{z}_{a_s} \cdot \boldsymbol{z}_{t_s})}{\sum_{t' \in B_{t_s}} \exp(\boldsymbol{z}_{a_s} \cdot \boldsymbol{z}_{t'_s}) + C^{t_d}} \tag{23}$$

Where $C^{t_d}$ the contribution of the concatenated samples to the above loss function.

$$C^{t_d} = \alpha_{same} \exp(\boldsymbol{z}_{a_d} \cdot \boldsymbol{z}_{\Pi(t)}) + \alpha_{diff} \sum_{t' \in B_{t_d} \setminus \{t\}} \exp(\boldsymbol{z}_{a_d} \cdot \boldsymbol{z}_{\Pi(t'_d)}) \tag{24}$$

The terms $\alpha_{same}$ in the above represent the concatenated samples which have one of the sounds similar to $\boldsymbol{z}_{a_s}$, while $\alpha_{diff}$ is the co–efficient used for all the concatenated samples ($\boldsymbol{z}_{a_d}$) which don't have any sound similar to $\boldsymbol{z}_{a_s}$. Next up we have similar formulation for the other half of the TeminAL A loss function which is shown below.

$$L_{a_A} = \sum_{(\mathbb{O}(a), \mathbb{O}(t)) \in B} (\text{TNCE}(\boldsymbol{z}_{a_s}, \boldsymbol{z}_{t_s}) + \text{TNCE}(\boldsymbol{z}_{a_d}, \boldsymbol{z}_{t_d})) \tag{25}$$

Finally the overall loss function for TeminAL A is composed of $L_{t_A}$ and $L_{a_A}$ shown as follows. Note, We keep all our hyper–parameters set as unity for the training of TeminAL A.

$$L_A = L_{t_A} + \beta_A(L_{a_A}) \tag{26}$$

The rest of the formulation follows the same derivation scheme as what we have detailed for TeminAL B in the above paragraphs.

### B.4 ZERO SHOT DOWNSTREAM TASK AND DETAILS:

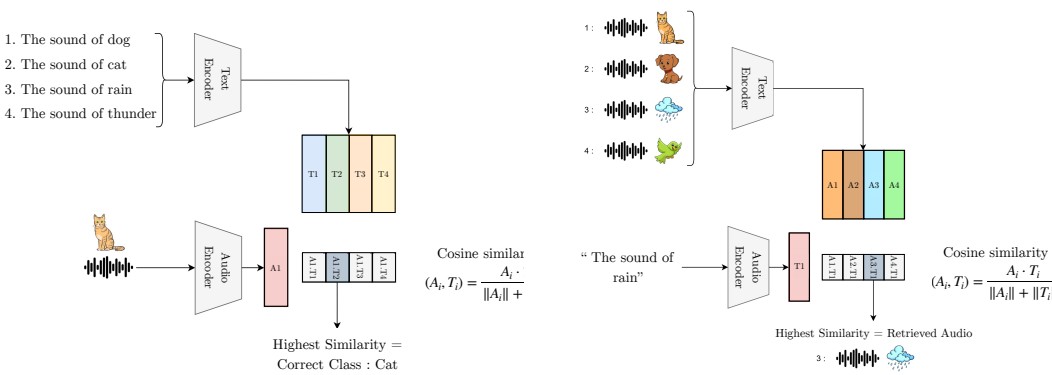

Figure 8: Zero Shot Audio Classification          Figure 9: Zero Shot Audio Retrieval

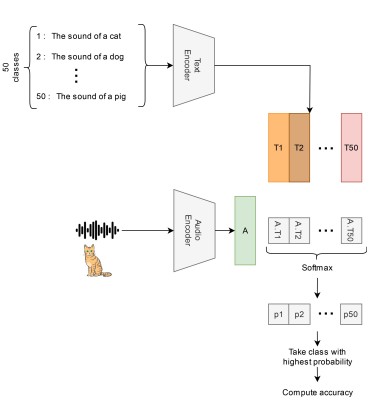

Figure 10: Configuration of task 1

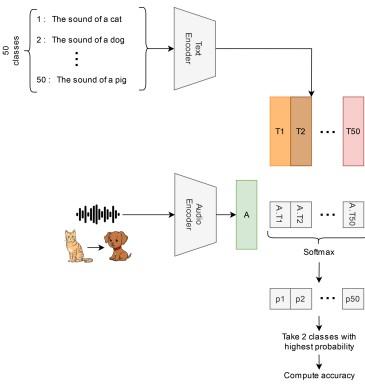

Figure 11: Configuration of task 2

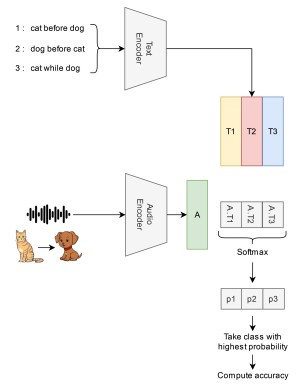

Figure 12: Configuration of task 3

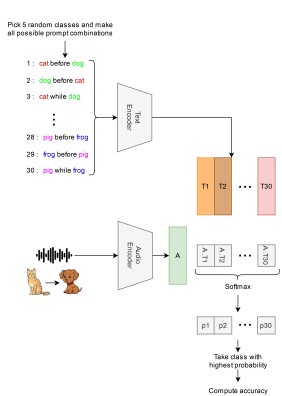

Figure 13: Configuration of task 4

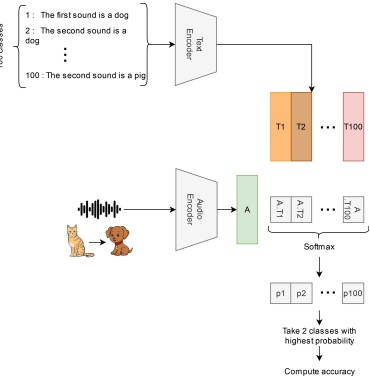

Figure 14: Configuration of task 5

Table 4: Performance comparison on audio classification task on different datasets. For ESC-50 and US8K we have used the prompt "The sound of a {class}" over all the 50 and 10 classes respectively. For ESC–50 the other text prompts are from the validation set of the model.

| Method | ESC-50 | US8K |
|---|---|---|
| Wav2CLIP | 41.4 | 40.4 |
| AudioClip | 69.4 | 65.3 |
| CLAP | 82.6 | 73.2 |
| CLAP-LAION-audio-630K | 88.0 | 75.8 |
| CompA-CLAP | 89.1 | 85.7 |
| T-CLAP (ours) | 75.1 | 72.2 |

---

**Algorithm 2 ZSTE**: **Z**ero **S**hot **T**emporal **E**valuation; evaluating Zero-Shot Temporal Classification Capabilities for General-purpose contrastive training multi-modal models. Implementation of ZSTE in our study is detailed in appendix B.4 also refer appendix B.4.1 for detail on parameters.

---

1: **Input:** Dataset $\mathcal{D}$, Contrastive Learning-based Model $\mathcal{M}$
2: **Output:** Model evaluation scores for zero-shot tasks $\mathcal{S}$
3: **Initialization:**
4:  Load dataset $\mathcal{D}$ and the contrastive learning-based model $\mathcal{M}$
5: **Task 1: Basic Zero-Shot Evaluation**
6:  Evaluate model's zero-shot capabilities on basic classification tasks $\mathcal{T}_1$
7:  Measure accuracy by correct label identification for unseen classes refer algorithm 3: $\text{Acc}_1 = \frac{1}{|\mathcal{U}|} \sum_{i \in \mathcal{U}} \mathbf{1}(\hat{y}_i = y_i)$
8:  Record baseline zero-shot performance $\text{Acc}_1$
9: **Task 2: Zero-Shot with Overlapping Features**
10:  Test model's ability to discern overlapping or composite features $\mathcal{T}_2$
11:  Measure accuracy based on correct label predictions for unseen composite instances refer algorithm 3: $\text{Acc}_2 = \frac{1}{|\mathcal{C}|} \sum_{j \in \mathcal{C}} \mathbf{1}(\hat{y}_j = y_j)$
12:  Record and analyze performance degradation or improvement $\mathcal{S}_2$
13: **Task 3: Temporal Relationship Comprehension**
14:  Present model with unseen sequences $\mathcal{Q}$ to assess temporal relationship understanding $\mathcal{T}_3$
15:  Measure accuracy in identifying the correct order of events refer algorithm 3: $\text{Acc}_3 = \frac{1}{|\mathcal{Q}|} \sum_{k \in \mathcal{Q}} \mathbf{1}(\hat{o}_k = o_k)$
16:  Evaluate against known sequences to determine zero-shot temporal comprehension $\mathcal{S}_3$
17: **Task 4: Resistance to Irrelevant Features**
18:  Challenge model with unseen data $\mathcal{N}$ that includes irrelevant features $\mathcal{T}_4$
19:  Determine model's ability to ignore noise and focus on relevant zero-shot features: $\text{Acc}_4 = \frac{1}{|\mathcal{N}|} \sum_{l \in \mathcal{N}} \mathbf{1}(\hat{y}_l = y_l)$
20:  Assess confusion metrics and resilience to irrelevant data $\mathcal{S}_4$
21: **Task 5: Generalization to Novel Scenarios**
22:  Evaluate model's generalization to completely novel zero-shot scenarios $\mathcal{T}_5$
23:  Measure model's performance on tasks with new contexts or relationships refer algorithm 3: $\text{Acc}_5 = \frac{1}{|\mathcal{X}|} \sum_{m \in \mathcal{X}} \mathbf{1}(\hat{y}_m = y_m)$
24:  Test for understanding of complex temporal sequences and novel feature combinations $\mathcal{S}_5$
25: **Conclusion:**
26:  Compile and compare evaluation scores across all tasks $\mathcal{S} = \{\mathcal{S}_1, \mathcal{S}_2, \mathcal{S}_3, \mathcal{S}_4, \mathcal{S}_5\}$
27:  Determine model's strengths and weaknesses in zero-shot learning
28:  Provide insights into model's potential real-world applicability
29: **return** Compiled evaluation scores $\mathcal{S}$, insights, and potential applications

---

- **Task 1** : In our initial experiment, we aimed to evaluate T–CLAP's performance on a straightforward classification task devoid of a temporal dimension. Our goal was to determine if T–CLAP exhibited any improvement or loss of capabilities compared to CLAP in this domain. We conducted this experiment by presenting the model with 50 distinct prompts in the format "The sound of [class label]", with each prompt corresponding to a

class in the ESC dataset. We then measured accuracy by assessing how often the model correctly identified the label associated with a given audio input (refer to Figure 10).

- **Task 2** : Subsequently, we explored whether T–CLAP demonstrated enhanced abilities in discerning two distinct sounds within a given audio clip with one of the sounds being from the validation set. The task configuration paralleled that of Task 1, with the key difference being that the accuracy assessment was conducted on audio clips featuring either concatenated or overlapping sounds (refer to Figure 11). We measured two accuracy metrics: one based on the model correctly identifying the two highest probabilities corresponding to the correct labels and another based on the model selecting at least one correct class.

- **Task 3** : In contrast to the preceding task, which disregarded temporality, this new experiment focuses on assessing T–CLAP's capability to accurately discern classes with their respective temporal relationships. For this task, we presented the model with three prompts following the same format as those encountered during training: "[class label 1] before [class label 2]", "[class label 2] before [class label 1]", and "[class label 1] while [class label 2]" (see Figure 12) while picking the 2 classes similar to Task 2. By exposing the model to an audio featuring one of these three temporal combinations, we gauged its accuracy in correctly identifying the corresponding temporal relationship within each prompt.

- **Task 4** : This task represents a more challenging iteration of Task 3. Here, our objective is to challenge the model by introducing prompts that include additional class labels not present in the audio (Figure 13), aiming to create confusion for the model during the evaluation process.

- **Task 5** : In our final task, we aimed to push the model's boundaries by presenting it with a temporal prompt it had not encountered during training, assessing its ability to generalize to novel temporal inputs. Our hypothesis was rooted in the nature of the text encoder, T5; if T–CLAP had truly grasped the temporal nuances embedded in "before" and "while" prompts, it should demonstrate an understanding of temporality across various prompt formats. For testing its comprehension of the "before" temporal aspect, we provided the model with four prompts structured as follows: "In this concatenated sound" followed by "The first sound is [class label 1]", "The second sound is [class label 1]", "The first sound is [class label 2]" and "The second sound is [class label 2]" (refer to Figure 14). In each instance, there were two correct prompts, and we evaluated the model based on its ability to correctly identify the combination of two prompts out of the six possible options. The model received a score of 1 if it correctly identified both prompts and 0.5 if it identified only one.

  Regarding the "while" temporality, we presented the model with 50 diverse prompts of the form "Simultaneous sound of [class label 1] and [class label 2]." The model's task was to select the two correct prompts, considering the two correct classes in both possible orderings. The same reward function was applied, scoring the model based on its accuracy in identifying both correct prompts or one, as appropriate.

### B.4.1    PARAMETER LIST FOR ALGORITHM 3

$\mathcal{D}$ : Dataset used for evaluation, $\mathcal{M}$ : Contrastive learning-based model being evaluated, $\mathcal{S}$ : Model evaluation scores for zero-shot tasks, $\mathcal{T}_1$ : Basic classification tasks for zero-shot evaluation, $\mathcal{U}$ : Set of unseen classes in basic classification tasks, $\text{Acc}_1$ : Accuracy for basic zero-shot classification tasks, $\hat{y}_i$ : Predicted label for the $i$-th unseen class, $y_i$ : True label for the $i$-th unseen class, $\mathcal{C}$ : Set of unseen composite instances in overlapping features tasks, $\text{Acc}_2$ : Accuracy for zero-shot tasks with overlapping features, $\hat{y}_j$ : Predicted label for the $j$-th composite instance, $y_j$ : True label for the $j$-th composite instance, $\mathcal{S}_2$ : Performance evaluation for overlapping features tasks, $\mathcal{Q}$ : Set of unseen sequences in temporal relationship comprehension tasks, $\text{Acc}_3$ : Accuracy for zero-shot temporal relationship comprehension tasks, $\hat{o}_k$ : Predicted order for the $k$-th sequence, $o_k$ : True order for the $k$-th sequence, $\mathcal{S}_3$ : Performance evaluation for temporal relationship comprehension tasks, $\mathcal{N}$ : Set of unseen data including irrelevant features, $\text{Acc}_4$ : Accuracy for tasks involving irrelevant features, $\hat{y}_l$ : Predicted label for the $l$-th instance in irrelevant features task, $y_l$ : True label for the $l$-th instance in irrelevant features task, $\mathcal{S}_4$ : Performance evaluation for resistance to irrelevant features, $\mathcal{T}_5$ : Tasks for evaluating generalization to novel scenarios, $\mathcal{X}$ : Set of instances in novel scenarios, $\text{Acc}_5$ : Accuracy for generalization to novel zero-shot scenarios, $\hat{y}_m$ : Predicted label for the $m$-th instance in novel scenarios, $y_m$ : True label for the $m$-th instance in novel scenarios, $\mathcal{S}_5$ : Performance evaluation for generalization to novel scenarios

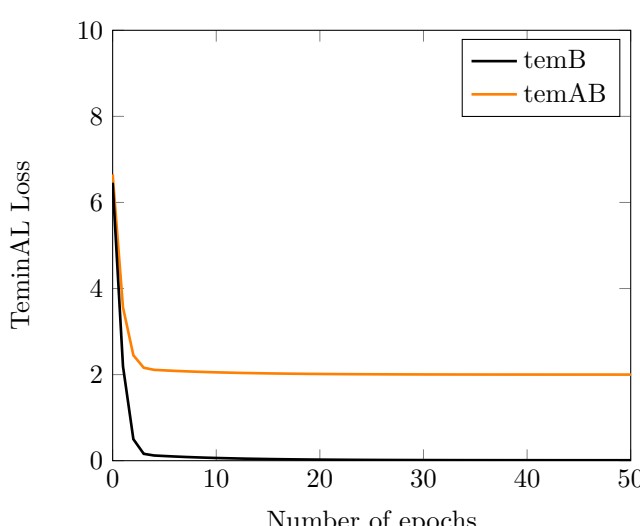

Figure 15: Model's performance of test dataset.

## B.5 TRAINING DETAILS

The model is trained on NVIDIA-RTX 4060 for 14hrs (including both TeminAL A and B). A total number of around 17.9 million parameters have been trained as detailed in table 5. The details of the dataset for the post–training is described in appendix B.2 while details of the training of the original CLAP model Elizalde et al. (2023) is shown in table 6. A total of 2450 audio–text pairs were used for TeminAL A and a total of 7350 audio-text pairs were used for the training of TeminAL B with a train–test split of 0.7. The batch size was selected as 256 after iterating on the size of 128, 256 and 512 as larger batches needed more iterations for convergence. Although we acknowledge that contrastive learning models generalises well for larger batch sizes as mentioned by Radford et al. (2021). The learning rate was chosen to be $10^{-4}$. Interestingly we found that the model trained only with TeminAL B converged but didn't do well on the learning as shown in fig. 15 due to the inability of the model to distinguish multiple sounds as explained in the section section 3.4. Thus the model warranted a hierarchical training with both TeminAL A and TeminAL B.

Table 5: Comparison of Text and Audio Parameters

| Parameter Type | Text | Audio |
|---|---|---|
| # Trainable Parameters | 9,515,520 | 8,423,951 |
| % of Total Parameters | 8.54% | 9.91% |

Table 6: Original model's (CLAP Elizalde et al. (2023)) training dataset statistics

| Dataset | Pairs | Unique audios | Unique captions |
|---|---|---|---|
| FSD50k | 36,796 | 36,796 | 36,796 |
| ClothoV2 | 29,646 | 5,929 | 29,646 |
| AudioCaps | 44,292 | 44,292 | 44,292 |
| MACS | 17,276 | 3,930 | 17,276 |
| Total | 128,010 | 90,947 | 128,010 |

### B.6 BASELINE MODELS

In evaluating retrieval tasks, specifically text-to-audio and audio-to-text, we assess CompA-CLAP alongside six other baseline models. MMT Oncescu et al. (2021) revolutionized the task of audio retrieval by introducing the use of free-form natural language queries, suggesting this method is more natural and versatile compared to traditional techniques reliant on text annotations. The research also highlights the advantages of pre-training on a variety of audio tasks. ML-ACT Mei et al. (2022) investigates the effects of distinct metric learning objectives on audio-text retrieval tasks, identifying the NT-Xent loss as a particularly effective method that consistently performs well across various datasets and training conditions, surpassing commonly-used triplet-based losses. Metric learning objectives are crucial for training cross-modal retrieval systems, as they organize data into an embedding space where similar items cluster together and dissimilar ones are separated. CLAP Elizalde et al. (2023) presents a new framework for retrieving audio utilizing a contrastive learning objective along with dual audio encoders to bridge the gap between language and audio content. Lastly, CLAP-LAION Wu et al. (2023b) offers a methodology for contrastive language-audio pre-training, aiming to forge robust audio representations by marrying audio data with corresponding natural language descriptions. Their model considers various audio and text encoders and enhances the model architecture with feature fusion strategies and keyword-to-caption augmentation.

| Model | T-A Retrieval | | | A-T Retrieval | | |
|---|---|---|---|---|---|---|
| | R@1 | R@5 | R@10 | R@1 | R@5 | R@10 |
| Pengi | 36.2 / 9.4 | 76.0 / 26.1 | 86.8 / 36.7 | 16.9 / 7.0 | 72.8 / 22.7 | 84.5 / 34.6 |
| Qwen-Audio | 39.1 / 16.2 | 78.9 / 45.8 | 87.1 / 57.2 | 38.0 / 16.1 | 73.2 / 23.3 | 85.0 / 35.1 |
| Audio Flamingo | **41.9 / 18.0** | **80.2 / 46.3** | **93.9 / 58.0** | 38.9 / 17.01 | 78.9 / 44.0 | 85.7 / 55.8 |
| CLAP | 34.6 / 16.7 | 70.2 / 41.1 | 82.0 / 54.1 | 41.9 / 20.0 | 73.1 / 44.9 | 84.6 / 58.7 |
| **T–CLAP(ours)** | 35.1 / 17.0 | 71.2 / 42.2 | 82.1 / 54.7 | **49.2** / 23.1 | **85.1 / 52.2** | 87.8 / 66.4 |

Table 7: Comparison of models with open-ended generation models on Text-Audio and Audio-Text retrieval performance on the AudioCap/Clotho dataset. The results for previous models have been taken from (Deshmukh et al., 2023; Elizalde et al., 2023). For retrieval in open-ended generation models, we use a consistent prompt style as mentioned in (Deshmukh et al., 2023).

### B.7 LIMITATIONS OF THE CURRENT MODEL

1. **General-Purpose ALM:** Our proposed model is not a general-purpose Audio-Language Model (ALM) capable of performing all downstream applications across all datasets. The current implementation is specifically designed and validated on the ESC-50 dataset as a proof-of-concept and to achieve our defined objectives. Consequently, generalization remains a limitation, although addressing this was beyond the scope of our work.

2. **Temporality Beyond the Dataset:** The model does not provide a general understanding of temporality beyond the ESC-50 dataset. Results from the ZSTE Task 4 and 5 confirm that neither does the model propose general temporality, nor does it achieve it. Notably, we emphasize that achieving general-purpose temporality would require larger, more comprehensive pre-trained text encoders. Specifically, open-ended text encoders (e.g., encoders from encoder-decoder models) would be more suitable than encoders trained on closed masked language modeling techniques, such as BERT.

3. **Zero-Shot Evaluation Scheme:** While our zero-shot evaluation scheme is designed to be general-purpose, it is inherently limited to contrastive models. Furthermore, the evaluation has not been tested on domains beyond the ESC-50 dataset. The reported ZSTE results are restricted to this dataset because it aligns with the training data used in our model and those of prior works. To mitigate data leakage, we ensure that our evaluation dataset is separate from the training data. However, it is important to note that the primary objective of this evaluation is to validate the proof-of-concept rather than to set new benchmarks.

4. **Evaluation Dataset Overlap:** A broader limitation, relevant to most models in this domain, is the overlap between evaluation datasets across various benchmarks. These overlaps can occur in terms of sound events or contextual similarities. Therefore, we cau-

tion against uncritical comparisons of model performance on classical benchmarks without careful consideration of dataset overlap.

By addressing these limitations, we hope to guide future researchers in expanding and improving upon the current work to achieve broader applicability and more generalized performance.

### B.8 EVALUATION METRICS

Our evaluation metrics are task specific, but in general they follow a similar strategy. The primary objective of the model appears to be to determine how well it can match audio clips with their corresponding textual descriptions. Here's a breakdown of key elements in the code and how they can be translated into a mathematical formulation for the evaluation section:

---

**Algorithm 3** General calculation for accuracy in ZSTE tasks

---

1: **Evaluation procedure**
2:     **Step 1: Audio Encoding**
3:         Encode audio inputs using the Audio Encoder $\mathcal{A}$ to get audio embeddings $\mathcal{A}_i$
4:         Ensure the embeddings are normalized to have a unit norm to maintain consistency in comparisons
5:     **Step 2: Similarity Calculation**
6:         Compute similarity scores between audio embeddings $\mathcal{A}_i$ and text embeddings $\mathcal{T}_j$ using a suitable similarity metric (e.g., cosine similarity)
7:         Generate a similarity matrix $\mathcal{S}$ where $\mathcal{S}_{ij}$ represents the similarity between the $i$-th audio embedding and the $j$-th text embedding
8:     **Step 3: Probability Calculation**
9:         Apply the softmax function to the similarity scores to obtain probabilities $p_{ij}$ for each class
10:         $p_{ij} = \frac{e^{\mathcal{S}_{ij}}}{\sum_{k=1}^{50} e^{\mathcal{S}_{ik}}}$
11:     **Step 4: Classification and Accuracy Measurement**
12:         Determine the predicted class by selecting the class with the highest probability for each audio input
13:         $\hat{y}_i = \arg\max_j p_{ij}$
14:         Measure accuracy by comparing predicted labels $\hat{y}_i$ with ground truth labels $y_i$: $\text{Acc}_1 = \frac{1}{|\mathcal{U}|} \sum_{i \in \mathcal{U}} \mathbf{1}(\hat{y}_i = y_i)$
15: **return** Evaluation scores $\text{Acc}_1$, insights, and potential improvements

---

1512
1513
1514
1515
1516
1517
1518
1519
1520
1521
1522
1523
1524
1525
1526
1527
1528
1529
1530
1531
1532
1533
1534
1535
1536
1537
1538
1539
1540
1541
1542
1543
1544
1545
1546
1547
1548
1549
1550
1551
1552
1553
1554
1555
1556
1557
1558
1559
1560
1561
1562
1563
1564
1565

---

**Algorithm 4** Audio-Text Matching Evaluation with CLAP Model

---

1: Initialize CLAP model with pre-trained weights
2: Load dataset $D = \{(a_i, t_i)\}_{i=1}^{N}$
3: Split dataset into training, validation, and test sets
4: Prepare DataLoader for batch processing
5: Load wordsList from file
6: Set prompt as 'this is a sound of '
7: Create target texts $y = [\text{prompt} + x \text{ for } x \text{ in words\_list}]$
8: **function** ONEHOTENCODE(text, wordsList)
9:     Initialize a zero vector $oneHotVector \in \{0, 1\}^{|\text{wordsList}|}$
10:     **for** each $word \in wordsList$ **do**
11:         **if** text starts with $word$ **then**
12:             Set $oneHotVector[\text{index of } word] \rightarrow 1$
13:             **break**
14:         **end if**
15:     **end for**
16:     **return** $oneHotVector$
17: **end function**
18: **for** each $batch \in testLoader$ **do**
19:     Extract $audio$ and $text$ samples from $batch$
20:     Compute audio embeddings $f_{\text{audio}}(a_i)$
21:     One-hot encode the text samples
22:     **for** each $t_j$ in text samples **do**
23:         $oneHotVector \leftarrow$ OneHotEncode($t_j$, wordsList)
24:         Compute text embeddings $f_{\text{text}}(oneHotVector)$
25:     **end for**
26:     Compute similarity scores $s(f_{\text{audio}}(a_i), f_{\text{text}}(\text{OneHotEncode}(t_j, \text{wordsList})))$
27:     Apply softmax to get $P(t_j|a_i)$
28:     Record predicted and true labels
29: **end for**
30: Compute accuracy:
31:     Accuracy $= \frac{1}{N} \sum_{i=1}^{N} \mathbb{I}(\hat{t}_i = t_i)$
32:     where $\hat{t}_i = \arg\max_{t \in T} s(a_i, t)$ and $\mathbb{I}$ is used as the indicator function.
33: **return** accuracy

---

