# OpenReview forum: "Enhancing Audio--Language Models through Self--Supervised Post--Training with Text--Audio Pairs"
_ICLR.cc/2025/Conference — Submitted to ICLR 2025_

### Official Review · Reviewer_xfBd · 2024-10-30

**Soundness:** 2
**Presentation:** 2
**Contribution:** 2
**Rating:** 3
**Confidence:** 5

**Summary:**

This paper presents a novel method for training audio-language contrastive models with sensitivity to temporal relations. This paper first constructed augmented data pairs that contain temporal relations, then trained with a novel 2-stage objective TeminAL AB. TeminAL AB first trains the model to distinguish single sounds and multiple sounds, then in the next stage it trains the model to distinguish between specific temporal relationships of the events. The objectives in TeminAL A and B consist of modified infoNCE loss which upweights the training of temporal relations. Last, this paper proposes a benchmark ZSTS to evaluate various audio-language zero-shot understanding tasks.

**Strengths:**

1. This paper proposes a novel 2-stage objective TeminAL AB for training audio-language contrastive models on temporal relationships.
2. This paper proposes a benchmark ZSTS to evaluate various audio-language zero-shot understanding tasks.

**Weaknesses:**

This paper lacks sufficient justification for its contributions, particularly due to limited comparison with prior works. Here are specific issues with the paper:

1. Comparison with Related Work: To my knowledge, the most relevant prior works are CompA [1] and T-CLAP [2]. The paper briefly introduces CompA in lines 90-104 but does not directly discuss how this work differs from or improves upon CompA.

2. Insufficient Comparative Analysis with CompA: CompA is only included for benchmarking performance. CompA generated and used a larger dataset with up to 110k samples, while this paper does not discuss the impact of dataset size or the types of datasets used in fine-tuning. CompA has its own method of defining tasks, such as using prompts like "before" and "after," while CompA uses "succeeded," "preceded," and "amidst." I am curious if the authors compared the effects of prompt design between these two models. Since CompA has a public benchmark, a fairer comparison would involve evaluating this model on CompA’s benchmark to highlight differences in approach. Perhaps the authors intend to show that their objective is superior to CompA’s, but there is no evidence provided to support this claim.

3. Comparison with T-CLAP: T-CLAP is also highly relevant, as both papers use the ESC-50 dataset to generate data. Although it seems the model and dataset in T-CLAP are not yet public, this work should at least include a discussion of T-CLAP’s approach.

4. Limitations in Contributions: Regarding the three contributions listed in this paper:
The first contribution has already been demonstrated in prior works [1-3].
The second requires more evidence and direct comparison with existing methods.
The third contribution lacks comparison with other benchmarks.

5. Discussion on Objective Functions: Although the authors note in the appendix that training with only the Terminal B objective is ineffective, I believe they should discuss why both Objectives A and B cannot be trained together.

Issues with Paper Presentation:
1. The text in the middle of Figure 3 is too small to read. I suggest placing Figures 3 and 4 together or at least referring to Figure 4 within Figure 3.
2. Line 334 contains a repeated "equation."
3. In line 334, it seems the numbering of the equation is incorrect. Eq 3 does not contain C^{cr} and C^{co}.

[1] Ghosh, S., Seth, A., Kumar, S., Tyagi, U., Evuru, C. K., Ramaneswaran, S., ... & Manocha, D. (2023). Compa: Addressing the gap in compositional reasoning in audio-language models. arXiv preprint arXiv:2310.08753.

[2] Yuan, Y., Chen, Z., Liu, X., Liu, H., Xu, X., Jia, D., ... & Wang, W. (2024). T-CLAP: Temporal-Enhanced Contrastive Language-Audio Pretraining. arXiv preprint arXiv:2404.17806.

[3] Wu, H. H., Nieto, O., Bello, J. P., & Salamon, J. (2023, June). Audio-text models do not yet leverage natural language. In ICASSP 2023-2023 IEEE International Conference on Acoustics, Speech and Signal Processing (ICASSP) (pp. 1-5). IEEE.

**Questions:**

See weakness.

---

> ### Author Response · Authors · 2024-11-16
>
> We are thankful for the insightful reviews and comments on our work. We are working on refining our subsmission and adressing all the questions marked in the review. Looking forward towards this discussion.
>
> Thanks,
> --Authors.

---

> ### Author Response · Authors · 2024-11-21
>
> We sincerely thank the reviewers for their thoughtful feedback, which has been invaluable in enhancing the quality of our paper. We deeply appreciate the recognition of our work's strengths and the constructive identification of areas for improvement. Below, we provide a detailed response to each identified weakness, address the specific questions raised, and outline the **changes incorporated into the revised submission**.
>
> We respectfully request the reviewers to carefully consider our explanations and arguments, as we believe they are integral to advancing the understanding we aim to contribute to the community through this work. We remain fully open to further discussions and are committed to refining the manuscript to better align with the expectations of the reviewers and the broader research community
>
> > **W1 \& W2 Citations and Comparisons:**
>
>    - The reviewer correctly points out the relevance of both [1] and [2], which we included as major contributions in this space. Additionally, we acknowledge that [3] is one of the earlier works in this domain. However, we did not include [3] in our original submission since it has neither been published in a peer-reviewed journal or conference nor has it made its code and results publicly available. We apologize for this oversight and have now cited [3] in the updated version of our paper.
>
>    - In response to the reviewer's comments, we have added a paragraph in section 1. There is indeed a distinct difference between our sequential learning approach and the modular contrastive approach mentioned by Compa [2]. While Compa trains the model to become modular by introducing progressively complex texts—essentially akin to introducing new types of labels in steps—our model adopts a fundamentally different strategy. We initially train the model to recognize that multiple distinct sounds can coexist, followed by training on these multiple sounds. This approach is more interpretable and structured in its methodology.
>
> - In comparison to T-CLAP and work by Wu et. al., both trains a contrastive learning model without requiring to address the need of multiple sounds distinction which defeats the purpose of increasing the interpretability of the models. We are not able to include them in the results section as they neither have benchmark results nor publicly available model trained weights for us to use. (Line 93-109).
>
> - Furthermore, In contrast, our approach achieves significant advancements within a limited computational budget, training around 10% of the total trainable parameters and utilizing a single dataset (ESC-50). Unlike prior works, such as those by Ghosh et al. (2023) and Wu et al. (2023a), which rely on more expansive datasets and substantial computational resources. Our focus is on developing a methodology that can effectively instill a sense of time in the model within acceptable computational constraints, rather than on generalizing over large, diverse dataset
>
>    **References:**
>    - [1] Wu et al., *Audio-text models do not yet leverage natural language*
>    - [2] Ghosh et al., *Compa: Model comparison*
>    - [3] *T-CLAP: Temporal-Enhanced Contrastive Language*
>
> > **W3. Public Accessibility of T-CLAP:**
>
> - As noted by the reviewer, T-CLAP [3] does not provide publicly available data or a codebase. This lack of public resources limits a direct comparison with their implementation. The details of coparison with this model has been included in line [96-99] and [100-106] the revised manuscript. We also like to correct the reviewer that T-CLAP [3] is trained on 2.04M audio--caption pairs and not just on ESC--50 dataset. While with our mdoel, the focus is on developing a methodology that can effectively instill a sense of time in the model within acceptable computational constraints, rather than on generalizing over large, diverse datasets.
>
> > **W4. Contributions:**
>
> Kindly check the following answer where we have detailed our contributions.
>
> > **W5. Two-Stage Training Methodology:**
>
>    The reviewer's comments on this aspect are highly insightful, and we have addressed this point in the updated manuscript.
>    - A joint training method would undermine the purpose of distinguishing between multiple sounds, which is a prerequisite for effective temporal instillation. Joint training enforces these conditions simultaneously, increasing the complexity for the model.
>    - Given the limited number of trainable parameters in our setup, we opted for a step-wise training approach, which is more controllable.
>    - Sequential training allows us to observe the model's progress at each stage, thereby enhancing interpretability.
>
> >  **Issues with the presentation :**
>
>    We are grateful for the reviewer's detailed observations and have made the following corrections in our updated submission:
>    - Updated Figure 3 and 4 for clarity.
>    - Removed the extra "equation"
>    - Corrected the equation numbering.

---

> ### Author Response · Authors · 2024-11-21
> **Explanation of our objectives, claims, and detailed proofs for each**
>
> **Overall Remark:**
>
> > **W4. Details about our contribution:**
>
> We would like to elaborate on our objectives, the claims associated with each, and the strong proofs presented as results. We kindly request the reviewers to thoroughly review these points, as they should provide a clearer understanding of the contributions and significance of our work.
>
> **Our Objectives (as presented in the paper):**
>
> > **1. Establishing a multi-stage temporal instillation objective:**
>
> The first stage focuses on learning to distinguish multiple sounds, while the subsequent stage emphasizes learning the temporal arrangement of these sounds.
>    - **Claim:** Training a single model to directly learn temporal relationships is ineffective because models must first develop the ability to distinguish between different sounds. Without this foundational understanding, learning temporality becomes infeasible.
>    - **Proof 1:** To validate this claim, we compared the model's performance after applying only TeminAL B with the performance after employing the multi-stage training process (TeminAL A followed by TeminAL B). The results show that the multi-stage model significantly outperforms in all tasks defined by ZSTE—a more advanced progression of the classical zero-shot evaluation strategy widely used in prior work. The results are detailed in Table 2.
>    - **Proof 2:** We further compared our model with state-of-the-art (SOTA) contrastively trained ALMs across all ZS evaluation tasks. Specifically, in ZSTE Task 2 (subtasks A and C), which require distinguishing multiple sounds, our model outperforms models trained on large datasets. However, in subtasks 2B and 2D (which involve simpler mappings of individual sounds), models trained on massive datasets excel due to their extensive exposure. For Tasks 3A and 3B, all models perform comparably, highlighting the challenges of zero-shot evaluations when trained on similar examples. These findings underscore the necessity of multi-stage training for better generalization.
>    - **Note:** Our training was intentionally restricted to a subset of the total parameters and used only the ESC-50 dataset. This decision aligns with our goal of demonstrating results within a realistic compute budget. Reviewers are kindly requested to consider that the focus of this work is not on achieving SOTA performance across billion-sample datasets, but rather on addressing a critical gap in current ALM post-training methods. Our key contributions lie in introducing a novel multi-stage training strategy and an effective evaluation framework.
>
> >**2. Demonstrating correct temporal relationships between audio and text:**
>
>    - **Claim:** Our analysis indicates that current contrastive ALMs face challenges in accurately capturing temporal relationships between audio and text.
>    - **Proof:** This claim is best evaluated after understanding our first objective. We acknowledge that the presentation order of objectives in the manuscript might have caused confusion, and we apologize for any inconvenience caused. However, the current order was chosen to maintain the logical progression of ideas.
>
> >**3. Proposing a general-purpose zero-shot evaluation scheme:**
>
>    - **Claim:** We are not claiming anything here, but providing and using a structured way of evaluating our earlier claims.
>    - **Proof:** Existing evaluation metrics fall short of effectively assessing temporal relationships, particularly in our case. We propose this evaluation scheme as a simple yet intuitive method to capture model behavior accurately. While future enhancements to evaluation metrics are certainly necessary, this scheme is sufficiently robust for our purposes.
>
> **Final Note:**
>
> If the addressed weaknesses and clarifications satisfactorily resolve the reviewers' concerns, **we kindly request an updated score** to reflect the potential impact of this work in advancing the field of audio-language models. Thank you.

---

> > ### Comment · Reviewer_xfBd · 2024-11-25
> >
> > I appreciate the authors’ detailed responses to the initial review. I recognize that the key distinction of this work, compared to existing research on training CLAP-like models for temporal relationships, is its focus on enabling such models to effectively distinguish between multiple sounds. However, the main issue with this paper remains the presentation and justification of its claims of contribution. Specifically, the paper does not sufficiently introduce the current state of the field or clearly delineate its distinctions and improvements over existing methods. I recommend the authors be more explicit and precise in describing their contributions relative to prior work. Below are the key points for further consideration:
> >
> > Distinction from Related Work
> > The paper should provide a clearer distinction from existing works on training CLAP-like models for temporal relationships, particularly the published works [1–3]. In Section 2.1, the related works section mentions CompA:
> > "Frameworks like CLAP and Compa (Elizalde et al., 2023; Ghosh et al., 2023) unify auditory-linguistic domains, offering strong zero-shot performance in multimodal tasks."
> > This discussion merely notes that CompA builds on CLAP to improve compositional reasoning, including temporal compositionality. Given the existence of these prior works, the authors should emphasize the distinction of their approach more explicitly. This distinction is currently buried within the tasks described in the ZSTS benchmark. Furthermore, the importance of distinguishing multiple sounds for temporal reasoning should be elaborated upon with detailed discussion and validation.
> >
> > Claims About Existing Models
> > The paper implies that existing models are incapable of capturing temporal relationships, which is not entirely accurate. For example, the authors state:
> > "We demonstrate that current contrastive ALMs fail to capture correct temporal relationships between audio and text, as shown in Table 3, revealing a critical gap in existing models."
> > However, Table 3 and the authors’ own responses indicate a more nuanced picture:
> > "Specifically, in ZSTE Task 2 (subtasks A and C), which requires distinguishing multiple sounds, our model outperforms models trained on large datasets. However, in subtasks 2B and 2D (which involve simpler mappings of individual sounds), models trained on massive datasets excel due to their extensive exposure. For Tasks 3A and 3B, all models perform comparably, highlighting the challenges of zero-shot evaluations when trained on similar examples."
> > Tasks 2 and 3 evaluate temporal relationships, and the results do not substantiate the claim that current models entirely fail at this. For example, CompA performs well in these tasks except for 2B and 2D. The authors should revise such sweeping statements to more accurately reflect the evidence.
> >
> > Comparison and Fairness in Evaluation
> > While I understand the computational differences between this work and [1–3], the comparison between this model (partially fine-tuned on ESC-50) and CompA (trained on larger datasets) lacks fairness. The authors acknowledge that models trained on larger datasets perform better in certain subtasks due to their extensive exposure, as stated in their response:
> > "However, in subtasks 2B and 2D (which involve simpler mappings of individual sounds), models trained on massive datasets excel due to their extensive exposure."
> > To address this, I recommend conducting a fair comparison by either training CompA using the dataset in this paper (with CompA’s original objective), or testing the proposed model on CompA’s benchmark.
> > As also noted by other reviewers, there is concern that this model may be overfitting to ESC-50, which warrants further investigation.
> >
> > Joint Training and Ablations
> > I remain unconvinced by the authors’ response regarding the exclusion of joint training results. At the very least, this should be included as an ablation test to strengthen the empirical validation.
> >
> > Conclusion:
> > For the reasons outlined above, I will retain my current score. The paper has potential but requires significant revision to address issues in presenting its contributions, making fair comparisons, and justifying its claims.
> >
> > References:
> > [1] Ghosh, S., Seth, A., Kumar, S., Tyagi, U., Evuru, C. K., Ramaneswaran, S., ... & Manocha, D. (2023). Compa: Addressing the gap in compositional reasoning in audio-language models. arXiv preprint arXiv:2310.08753.
> >
> > [2] Yuan, Y., Chen, Z., Liu, X., Liu, H., Xu, X., Jia, D., ... & Wang, W. (2024). T-CLAP: Temporal-Enhanced Contrastive Language-Audio Pretraining. arXiv preprint arXiv:2404.17806.
> >
> > [3] Wu, H. H., Nieto, O., Bello, J. P., & Salamon, J. (2023, June). Audio-text models do not yet leverage natural language. In ICASSP 2023-2023 IEEE International Conference on Acoustics, Speech and Signal Processing (ICASSP) (pp. 1-5). IEEE.

---

> > > ### Author Response · Authors · 2024-11-25
> > >
> > > Respected Reviewer,
> > >
> > > We thank you for the detailed articulation of your concerns. We have prepared a well-structured overview of our responses and will provide comprehensive replies to each point. However, we would like to humbly note that some of the concerns raised seem repetitive and do not fully engage with the arguments we previously presented. This suggests that the issue may lie in the clarity of our presentation, which we believe is entirely solvable and the onus is on us to rectify it and present our ideas meaningfully. Rest assured, we are committed to addressing these issues thoroughly and thoughtfully.
> > >
> > > **We greatly appreciate the reviewer’s positive recognition of the potential of this work and are dedicated to enhancing the context and presentation of our contributions.** We will submit our detailed response and an updated draft at the earliest opportunity, ensuring the reviewer has enough time to review and provide further feedback. Thank you once again for your time and effort in helping us improve this work.
> > >
> > > Sincerely,
> > >
> > > --Authors.

---

> ### Author Response · Authors · 2024-11-25
>
> > **W1: Distinction from other Related work** : We request the reviewer to kindly check the updated document where we have specifically updated the following to inlcude all the distinction between current and related works. **Kindly note the specific lines**.
>
>    - **(Line 93-101)** : In response to the reviewer's comments, we have added a paragraph in section 1. There is indeed a distinct difference between our sequential learning approach and the modular contrastive approach mentioned by Compa [2]. While Compa trains the model to become modular by introducing progressively complex texts—essentially akin to introducing new types of labels in steps—our model adopts a fundamentally different strategy. We initially train the model to recognize that multiple distinct sounds can coexist, followed by training on these multiple sounds. This approach is more interpretable and structured in its methodology.
>
> - **(Line 102-108)** : Furthermore, In contrast, our approach achieves significant advancements within a limited computational budget, training around 10% of the total trainable parameters and utilizing a single dataset (ESC-50). Unlike prior works, such as those by Ghosh et al. (2023) and Wu et al. (2023a), which rely on more expansive datasets and substantial computational resources. Our focus is on developing a methodology that can effectively instill a sense of time in the model within acceptable computational constraints, rather than on generalizing over large, diverse dataset
>
> -  **(Line 97-101) and (Line 101-104)** :  In comparison to T-CLAP and work by Wu et. al., both trains a contrastive learning model without requiring to address the need of multiple sounds distinction which defeats the purpose of increasing the interpretability of the models. We are not able to include them in the results section as they neither have benchmark results nor publicly available model trained weights for us to use.
>
> - **(Section 5 ; Specifically in discussion to Table 2 and Table 3)** : Non explicit comparison based on model results, the reviewer has already noted these comparative details as mentioned in their comment.
>
> > **W2. Claim on current model** : Yes, as mentioned in our previous comment.
>
> - This claim is best evaluated after understanding our first objective. We acknowledge that the presentation order of objectives in the manuscript might have caused confusion, and we apologize for any inconvenience caused. However, the current order was chosen to maintain the logical progression of ideas.
>
> - As mentioned by us it is clear from the result section that the models trained on larger dataset with large number of trainable paramters do perform well in subtask 2B and 2D, as it only requires them to map to at--least one of the sounds (There was a type-o in our initial response, we corrected it.). Now when we take a look at subtask 2A and 2C, it is evident that the other models do not satisfactorily capture the distniguishing multplie sounds.
>
> - We thank the reviewer for taking this issue up we accept the suggestion by the reviewer and to address the issue which the reviewer has, we have updated the main paper **(Line 131-133)** : " Our analysis indicates that current contrastive ALMs face challenges in accurately capturing temporal relationships between audio and text, as shown in \cref{table5}, highlighting an area for potential improvement in existing models." We request the author to kindly review this and let us know if this addresses the concerns on our remarks given to the previous models.

---

> ### Author Response · Authors · 2024-11-25
>
> > **W3. Comparison and Fairness of evaluation** :
>
> - We thanks the reviewer for noting the impact of our post--training dataset size and the parity with other mentioned models. We acknowledge the reviewers suggestion of either training CompA on the same dataset or using CompA's dataset for comparison. To carefully adress this we have the following remarks.
>
> - > **Results overfitting on ESC--50:**
>
> - We do partially accept this remark from the reviewer and think it's an important remark in general toward model post--training. Although, as we emphasis in our objective and scope of work, that our aim is not give a generalisable model which perform SOTA across all datasets. Rather our aim is to present the idea of multi--stage training of contrastive models and it should effectively be done in the sense of instillation of time. We can't really say it's overfitting as we perform all our multiple audio evaluation on the test set for ESC--50. Although we do accept that the model hasn't seen other data and so has a bias towards ESC--50 which does get reflected in the results. But given the compute budget, we determine training on a large dataset to be out of scope for the current work.
>
> - > **Assertion:** Evaluating results of other mentioned dataset would not be able to give better result than the origianl CLAP model, as we never post--trained on those specific datasets. Henceforth encountering any OOD sample would mean feeding unkown data to the model to hallucinate on. While the training can well be extended to different datasets or to a better mixture of datasets, but that defeats the purpose of being able to train within our compute budget. Although this point has been well noted for future work which will emphasis solely on giving generalisable post--trained models. **We request the reviewer to share their view on our assertion**.
>
> - Thus, our final take on this is to not consider the model to be a generalisable SOTA ALM, rather consider the method to be as one which has novelty in contribution and succesfully defends the novelty as detailed in our previous comment (https://openreview.net/forum?id=nplYdpc1Pm&noteId=e4ylyCLZy1) without going out of scope. So, it's funadmentally neither fair nor reasonable to evaluate the model on various datasets testing it's generalisability.
>
> > **Comment on joint training method:**
>
> - We would like to humbly counter the point mentioned by the reviewer, why are we focusing on the joint training mehtod? It doesn't take away from the fact that a multi-objective (either joint or sequential) is required.
> - In this work we opted for the sequential training method as we wanted to focus on understandibilty of each of our models.
> - The abalation on joint training would be a good addition, but it would not add knowledge to the eixting proofs of our claims. Our work objective is not to test the various training strategy which can be there, rather our objective is to assert that the current contrastive ALMs do not satisfatory capture the temporal instillation and for that we propose a method.
> - Henceforth, we humbly request the reviewer to kindly view the current objectives as it would be out of our scope to tackle all the challenges in one work.
>
> **Final note and request:** We sincerely appreciate the reviewer for taking the time to provide highly valuable comments on our responses. We have made all the necessary changes as requested by the reviewer. We request the reviewer to kindly review our updates and if they are able to convey the message then kindly update their scores.

---

> ### Author Response · Authors · 2024-11-28
> **Final update**
>
> Respected reviewer xfBd,
>
> Along with the above updates on your questions, we have taken the time to adress all the reviews and provide an in-depth detail of the summary (The response can be found up top in the comment section as **"Final updates and addressing the reviewers"** , https://openreview.net/forum?id=nplYdpc1Pm&noteId=tGw3QZiByT, Detailed paper review: https://openreview.net/forum?id=nplYdpc1Pm&noteId=lfCYvH8OhT) along with the limitations (https://openreview.net/forum?id=nplYdpc1Pm&noteId=jj9j5D2lq6) of our work. Kindly review this and if it satisfactorily answers your concerns then reward us with a better score.
>
> Thanks.
> --Authors.

---

> > ### Author Response · Authors · 2024-12-03
> > **Final day of review (2nd Dec)**
> >
> > Respected Reviewer xfBd,
> >
> > We have carefully addressed all the concerns you mentioned, particularly regarding the limitations you highlighted. We humbly request you to kindly review the updates provided above, along with our responses to your previous two comments, as they directly address your concerns.
> >
> > We greatly appreciate your time and effort and respectfully request your consideration for a more favorable score.
> >
> > Thank you.

---

> ### Author Response · Authors · 2024-12-03
> **Final request (2nd Dec, within time)**
>
> Respected reviewer,
>
> If you don't have any further arguments related to your queries, kindly update your score. If you do have arguments or queries, please post your rebuttal so that we can address them. **We respect your position of responsibility and time, and humbly expect the same courtesy from your end.**
>
> From our side, we have addressed all your concerns and accepted all your insightful suggestions. However, on your end, we expected you to either acknowledge or update the score.
>
> Thanks.

---

> > ### Author Response · Authors · 2024-12-03
> > **Buffer day request (3rd Dec, within score update time)**
> >
> > Respected Reviewer,
> >
> > I hope this message finds you well. We have made a sincere effort to address all the concerns raised in your feedback to the best of our ability, ensuring that all limitations are clearly articulated. We kindly request you to consider updating the score in light of these revisions.
> >
> > Having worked meticulously on the revisions for over a month, we would greatly appreciate it if you could spare some time today to review them, as the opportunity for score updates remains available. Your thoughtful consideration would mean a great deal to us.
> >
> > Thank you for your time and understanding.
> >
> > Best regards,
> >
> > --Authors

---

### Official Review · Reviewer_SKUo · 2024-11-02

**Soundness:** 3
**Presentation:** 2
**Contribution:** 3
**Rating:** 5
**Confidence:** 4

**Summary:**

The authors propose to include both data curation and loss function design in helping the CLAP model to understand better the temporal relation. They also propose a zero shot temporal evaluation framework with a series of tasks in order to benchmark specific temporal modeling capability.

**Strengths:**

- Temporal modeling capability in audio is a fairly novel and under-explored research area. The proposed zero shot evaluation benchmark is reasonable and can be a good contribution to the community.

**Weaknesses:**

- The use of equations in section 3 to explain the proposed loss function is not easy and straightforward to follow, and they do not connect well with figure 3. Consider adding some of the variables, loss terms from the equation into figure 3 to improve the connection for clearer explanation.
- TeminAL A can benefit from more detailed explanation. It is mentioned that TeminAL A is trained to distinguish between single and multiple sounds,  are they trained with similar contrastive loss? if so, how are the audio and text pairs curated? or are they trained with classification loss separately? It might worth adding a figure to explain the TeminAL A stage training, similar to figure 4 for TeminAL B.
- The more detailed explanation of ZSTE should be included in the main paper, current appendix B.4 contains most of the information for the reader to understand what proposed ZSTE is? Since these are one of the main contributions of this work, it is worth integrate into the main narratives.
- The explanation of the categories A, B, C, D used in Table 2 is buried in the caption of Table 3, these are also important information and might worth moving into the narrative in the sections explaining the evaluation and benchmarks.
- Minor suggestion: consider making the fonts in the figures larger, there are still space in the figure that can be adjusted.
- For section 3.3, in the l_text, x_c and x_a are referred to the raw inputs, if the loss function takes embeddings as inputs, should these variables be z instead of x?

**Questions:**

- In table 1, are the results from different model in comparison trained by the authors? Or are they taking from publicly available models from each work?

---

> ### Author Response · Authors · 2024-11-16
>
> We are thankful for the insightful reviews and comments on our work. We are working on refining our subsmission and adressing all the questions marked in the review. Looking forward towards this discussion.
>
> Thanks,
> --Authors.

---

> ### Author Response · Authors · 2024-11-20
>
> We sincerely thank the reviewers for their thoughtful feedback, which has been invaluable in enhancing the quality of our paper. We deeply appreciate the recognition of our work's strengths and the constructive identification of areas for improvement. Below, we provide a detailed response to each identified weakness, address the specific questions raised, and outline the **changes incorporated into the revised submission**.
>
> We respectfully request the reviewers to carefully consider our explanations and arguments, as we believe they are integral to advancing the understanding we aim to contribute to the community through this work. We remain fully open to further discussions and are committed to refining the manuscript to better align with the expectations of the reviewers and the broader research community
>
> **Weaknesses:**
>
> > **W1. Title of Figure 3:**
>
> Thank you for pointing this out. We accept this comment and have modified the title of Figure 3 for clearer message delivery. Since Figure 3 serves as an overview of the model architecture, we have provided an in-depth explanation of the loss function in Figures 6 and 7. Additionally, we have included references to these figures in Section 3.4 to improve ease of understanding and relatability.
>
> > **W2. Loss Function Formulation for TeminAL A:**
>
> We acknowledge the reviewer's suggestion and have included a figure demonstrating the loss function formulation for TeminAL A in the relevant section. Due to space constraints, we have placed detailed derivations in the appendix, in line with current trends in conference papers. However, key equations are retained in the main section, starting from line 366, including Equations 8 and 9. Further derivations are provided in the supplementary material from Equation 22 onward. Morever we have also included Figure 7 for reference to understand TeminAL A loss function.
>
> > **W3 \& W4. Highlighting ZSTE in the Main Paper:**
>
> We appreciate the suggestion to better highlight ZSTE in the main paper. The whole Section 4.2 has been revised to incorporate necessary information from Appendix Section B.4. Due to space limitations, we have moved some background details and additional diagrams to the supplementary section. Nonetheless, all important impacts and applications of ZSTE are now detailed in the Results section (Section 5) as we have re-written the whole section. The results section includes Table 1, Table 2, and explanatory paragraphs in Section 5, which specifically highlight the role of ZSTE in our current work.  The details of ZSTE are now shifted to main text from the tables as per suggestion.
>
> > **W5. Figure Fonts:**
>
> The font size of the figures has been increased, following the reviewer's helpful suggestion.
>
> > **W6. Clarity on Embeddings and Variables:**
>
> We agree with the reviewer's observation regarding the embeddings going into the loss functions, denoted as $z$, and the formulation of $l_{\text{text}}(C)$. These aspects have been clarified to ensure better understanding of the variable usage.
>
> **Questions:**
>
> > **Q1. Source of the results from table 1**
>
> **Answer** Yes, the results of the model in Table 1 have been taken from previous published results and the sources have been cited. The citations have been included the caption of the figure as well.
>
> We thank the reviewer again for the insightful comments, if all the parts of the weknesses and the questions have been adressed well in our modifications , we request the reviewer to kindly update the score of our paper. In order to mark the impact of the work for the development of such models.

---

> ### Author Response · Authors · 2024-11-20
> **Overall comment on the explanation of our objectives, claims, and detailed proofs for each**
>
> **Overall Remark:**
>
> We would like to elaborate on our objectives, the claims associated with each, and the strong proofs presented as results. We kindly request the reviewers to thoroughly review these points, as they should provide a clearer understanding of the contributions and significance of our work.
>
> **Our Objectives (as presented in the paper):**
>
> > **1. Establishing a multi-stage temporal instillation objective:**
>
> The first stage focuses on learning to distinguish multiple sounds, while the subsequent stage emphasizes learning the temporal arrangement of these sounds.
>    - **Claim:** Training a single model to directly learn temporal relationships is ineffective because models must first develop the ability to distinguish between different sounds. Without this foundational understanding, learning temporality becomes infeasible.
>    - **Proof 1:** To validate this claim, we compared the model's performance after applying only TeminAL B with the performance after employing the multi-stage training process (TeminAL A followed by TeminAL B). The results show that the multi-stage model significantly outperforms in all tasks defined by ZSTE—a more advanced progression of the classical zero-shot evaluation strategy widely used in prior work. The results are detailed in Table 2.
>    - **Proof 2:** We further compared our model with state-of-the-art (SOTA) contrastively trained ALMs across all ZS evaluation tasks. Specifically, in ZSTE Task 2 (subtasks A and C), which require distinguishing multiple sounds, our model outperforms models trained on large datasets. However, in subtasks 2B and 2D (which involve simpler mappings of individual sounds), models trained on massive datasets excel due to their extensive exposure. For Tasks 3A and 3B, all models perform comparably, highlighting the challenges of zero-shot evaluations when trained on similar examples. These findings underscore the necessity of multi-stage training for better generalization.
>    - **Note:** Our training was intentionally restricted to a subset of the total parameters and used only the ESC-50 dataset. This decision aligns with our goal of demonstrating results within a realistic compute budget. Reviewers are kindly requested to consider that the focus of this work is not on achieving SOTA performance across billion-sample datasets, but rather on addressing a critical gap in current ALM post-training methods. Our key contributions lie in introducing a novel multi-stage training strategy and an effective evaluation framework.
>
> >**2. Demonstrating correct temporal relationships between audio and text:**
>
>    - **Claim:** Our analysis indicates that current contrastive ALMs face challenges in accurately capturing temporal relationships between audio and text.
>    - **Proof:** This claim is best evaluated after understanding our first objective. We acknowledge that the presentation order of objectives in the manuscript might have caused confusion, and we apologize for any inconvenience caused. However, the current order was chosen to maintain the logical progression of ideas.
>
> >**3. Proposing a general-purpose zero-shot evaluation scheme:**
>
>    - **Claim:** We are not claiming anything here, but providing and using a structured way of evaluating our earlier claims.
>    - **Proof:** Existing evaluation metrics fall short of effectively assessing temporal relationships, particularly in our case. We propose this evaluation scheme as a simple yet intuitive method to capture model behavior accurately. While future enhancements to evaluation metrics are certainly necessary, this scheme is sufficiently robust for our purposes.
>
> **Final Note:**
>
> If the addressed weaknesses and clarifications satisfactorily resolve the reviewers' concerns, **we kindly request an updated score** to reflect the potential impact of this work in advancing the field of audio-language models. Thank you.

---

> > ### Comment · Reviewer_SKUo · 2024-11-25
> >
> > Thanks to the authors for updating the script and make it more clear in the necessary explanation for the loss function and downstream tasks. I am going to increase the rating.

---

> ### Author Response · Authors · 2024-11-25
>
> Respected Reviewer,
>
> We sincerely thank the reviewer for their thoughtful feedback and guidance, which have greatly contributed to improving our work. We appreciate your acknowledgment of the updates made in response to your earlier comments. We kindly request additional feedbacks on how we can further refine our manuscript to address your concerns and ensure it aligns with the standards for acceptance. Based on the current score, we feel that we may not have effectively communicated the key contributions and strengths of our work.
>
> Furthermore, **if our efforts to address your concerns regarding novelty, the proofs provided, and the discussions in the paper have met your expectations** and if the reviewer thinks the contribution in this work is novel and should be highlighted, then it's a humble request to score this work towards acceptabiity and help us share this knowledge towards the general development of the scientific community.
>
>
> With gratitude,
>
> --The Authors

---

> ### Author Response · Authors · 2024-11-28
> **Final update**
>
> Respected reviewer SKUo,
>
> Along with the above updates on your questions, we have taken the time to adress all the reviews and provide an in-depth detail of the summary (The response can be found up top in the comment section as **"Final updates and addressing the reviewers"** , https://openreview.net/forum?id=nplYdpc1Pm&noteId=tGw3QZiByT, Detailed paper review: https://openreview.net/forum?id=nplYdpc1Pm&noteId=lfCYvH8OhT) along with the limitations (https://openreview.net/forum?id=nplYdpc1Pm&noteId=jj9j5D2lq6) of our work. Kindly review this and if it satisfactorily answers your concerns then reward us with a better score.
>
> Thanks.
> --Authors.

---

> > ### Author Response · Authors · 2024-12-03
> > **Final day for reviews (2nd Dec)**
> >
> > Respected Reviewer SKUo,
> >
> > We have carefully addressed all the concerns mentioned by the reviewers for improving our work, particularly regarding the limitations of our work. We humbly request you to kindly review the updates provided above in the detailed comment with links.
> >
> > We greatly appreciate your time and effort and respectfully request your consideration for a more favorable score.
> >
> > Thank you.

---

> > > ### Author Response · Authors · 2024-12-03
> > > **Buffer day request (3rd Dec, within the score update period)**
> > >
> > > Respected Reviewer,
> > >
> > > I hope this message finds you well. We have made a sincere effort to address all the concerns raised in your feedback to the best of our ability, ensuring that all limitations are clearly articulated. We kindly request you to consider updating the score in light of these revisions.
> > >
> > > Having worked meticulously on the revisions for over a month, we would greatly appreciate it if you could spare some time today to review them, as the opportunity for score updates remains available. Your thoughtful consideration would mean a great deal to us.
> > >
> > > Thank you for your time and understanding.
> > >
> > > Best regards,
> > >
> > > --Authors

---

### Official Review · Reviewer_isZu · 2024-11-02

**Soundness:** 2
**Presentation:** 2
**Contribution:** 3
**Rating:** 6
**Confidence:** 5

**Summary:**

This paper introduces an essential gap in contrastively pre-trained Audio-Language Models (ALMs) by showing their failure to capture correct temporal relationships between audio and various acoustic events. To address this, the authors propose TeminAL—a two-step post-training framework for infusing temporal awareness in ALMs. Additionally, the paper introduces ZSTE, a Zero-Shot Temporal Evaluation scheme designed to evaluate temporal understanding in ALMs in a zero-shot fashion.

**Strengths:**

1. The paper shows a critical gap in current ALMs for lacking temporal reasoning and acting as a "bag of words" while mapping audio and textual information using cosine similarity. To address this, the paper proposes Temin AL, a post-hoc alignment approach focusing on injecting temporal ordering among diverse acoustic events in existing ALMs
2. To evaluate temporal understanding in current ALMs in a zero-shot fashion, the authors also propose a multistage zero-shot temporal evaluation scheme, called ZSTE

**Weaknesses:**

1. The curriculum learning approach introduced by the authors while showing decent performance in audio-text retrieval tasks, still remains very similar to the CompA's modular contrastive pre-training approach. I will request the authors to address this in the main paper.
2. The authors have used ESC50 for synthesizing temporal-rich audio segments (e.g., dog after cat, etc.). Did the authors explore more diverse audio sources like AudioSet Strong, which contains time-aligned information and more than 500+ labels? I will request the authors to add a few lines in the paper to explain the rationale behind using ESC50 when compared to other audio data sources.
3. I find important ZS experiments on standard audio-classification tasks like AudioSet, FSD50K, USD8K, TUTUrban, etc. missing in the main paper.

**Questions:**

Additional Questions:
1. As shown in Figure 1, what is the advantage of 2-stage training vs 1stage training? Doing a comparative study will be useful in understanding which stage is contributing more towards temporal understanding.
2. The performance on standard order understanding benchmarks in audio like COMPA-Order and COMPA-attribute is missing in the main paper.

---

> ### Author Response · Authors · 2024-11-16
>
> We are thankful for the insightful reviews and comments on our work. We are working on refining our subsmission and adressing all the questions marked in the review. Looking forward towards this discussion.
>
> Thanks,
> --Authors.

---

> ### Author Response · Authors · 2024-11-20
>
> We sincerely thank the reviewers for their thoughtful feedback, which has been invaluable in enhancing the quality of our paper. We deeply appreciate the recognition of our work's strengths and the constructive identification of areas for improvement. Below, we provide a detailed response to each identified weakness, address the specific questions raised, and outline the changes **incorporated into the revised submission**. (Answer to the questions in the next comment, thanks)
>
> We respectfully request the reviewers to carefully consider our explanations and arguments, as we believe they are integral to advancing the understanding we aim to contribute to the community through this work. We remain fully open to further discussions and are committed to refining the manuscript to better align with the expectations of the reviewers and the broader research community.
>
> **Weaknesses:**
>
> > **W1. Comparison with Compa's modular contrastive training approach**
>
> Thank you for highlighting this point. There is indeed a distinct difference between our sequential learning approach and the modular contrastive approach mentioned by Compa. While Compa trains the model to become modular by introducing progressively complex texts—essentially akin to introducing new types of labels in steps—our model adopts a fundamentally different strategy. We initially train the model to recognize that multiple distinct sounds can coexist, followed by training on these multiple sounds. This approach is more interpretable and structured in its methodology. We have now included these comparison points in the main text (Line 93-109)
>
>    - Compa’s final objective remains the same across their training steps, and they do not provide a comparative analysis to demonstrate what the model gains in knowledge from the initial training on hard negatives. While they report percentage improvements in performance, they do not substantiate this with reasoning, which makes their method less interpretable.
>
>    - Unlike prior works, such as those by Ghosh et al. (2023) and Wu et al. (2023a), which rely on large datasets and substantial computational resources, our focus is on developing a methodology that instills a temporal sense within the model while remaining within realistic computational constraints. Our emphasis is on this methodological advancement rather than on generalization over expansive and diverse datasets.
>
> > **W2. Rationale behind using ESC50**
>
> We deeply appreciate this important point raised by the reviewer. We have addressed this in Section B2 (line 894 onwards) of our paper. As the work is authored by student researchers, it is important to recognize the implications of using a single dataset. Expanding the dataset would necessitate an increase in the model size (i.e., trainable parameters) to counteract the high bias introduced by increased dataset diversity. This would go beyond the scope and budget of our current work. Hence, we strategically focused on highlighting critical gaps in the current state of Audio-Language Model (ALM) research, particularly in multi-modal research, while adhering to realistic training constraints. We are immensely thankful to the reviewer for raising this point, as it underscores the deliberate choices we made to balance the scope and impact of our project.
>
> > **W3.  ZS experiments on standard audio-classification**
>
> We acknowledge the reviewer’s observation regarding the absence of standard audio classification results on datasets like FSD50K and USD8K. Our primary focus in this paper was to address temporal instillation and move beyond standard Zero-Shot (ZS) tasks. The objective of our work is to provide a meaningful step toward achieving these goals, as evidenced by our results compared to other SOTA models. That said, we accept the reviewer’s suggestion and have now included classification results from ESC-50 and USD8K in Table 4. We are also planning to incorporate results from FSD50K, contingent upon the timeline for submission. It is important to note that due to our focus on training with limited computational resources, we included only ESC-50 in our post-training dataset. As such, we rely on the original training of the CLAP model over these datasets, which explains the observed decrement in performance. However, our results demonstrate that the model does not undergo catastrophic forgetting, as evidenced by the competitive results on benchmark retrieval datasets such as Clotho and AudioCap.

---

> ### Author Response · Authors · 2024-11-20
>
> We thank the reviewer again for the insightful questions. We wish to answer the above questions in the following order. Also, if all the concerns have been adressed well, then we humbly request the reviewer to update the current score. Thank you :
>
> **Answers to the Questions:**
>
> > **Q1. what is the advantage of 2-stage training vs 1stage training?**
>
> **Answer.** The reviewer has correctly noted the impact of our two-stage training approach. To address this, we have included results from both the single-stage (TerminAL B only) model and the two-stage (TerminAL A + B) model in Tables 2 and 3. These results clearly demonstrate the contribution of Stage A training to the overall performance of the Stage B model. Specifically, as shown by the results, we validate our hypothesis that the model struggles to distinguish between multiple audio sources when trained with only Terminal B. We kindly request the reviewer to refer to Table 3, where we present comparative results for the model trained without Terminal A (i.e., without prior knowledge of multiple sound distinctions) and the model trained with Terminal A + B (where Terminal A knowledge is first introduced). Our observations confirm that the two-stage model exhibits a better understanding of temporality across various ZSTE tasks, substantiating the importance of our proposed methodology.
>
> >**Q2. COMPA-Order and COMPA-attribute missing in the main text**
>
> **Answer** The reviewer rightly points out the absence of zero-shot metrics like Compa-Order, Compa-Attribute (Ghosh. et. al.), and similar evaluations. Previous studies (Wu. et. al.) have demonstrated that existing zero-shot evaluation methods often fail to capture the general understanding of models comprehensively. While Compa includes such observations and establishes similar ZS metrics with slight variations in prompt design (e.g., Compa Order and Compa Attribute evaluations), their focus is primarily on assessing the model's general-purpose zero-shot capabilities.
>
>    In contrast, our current work introduces an improved evaluation metric that not only encompasses but also broadens the scope of these ZS metrics to provide more generalized and meaningful evaluation strategies. Additionally, we would like to highlight that we are also working on follow-up research to further enhance the generalizability of models beyond the scope of the current work. However, we believe that this paper represents a critical first step toward achieving more impactful and robust evaluations of general intelligence, and we appreciate the opportunity to advance the discourse in this area.

---

> ### Author Response · Authors · 2024-11-20
> **Overall comment on the explanation of our objectives, claims, and detailed proofs for each**
>
> **Overall Remark:**
>
> We would like to elaborate on our objectives, the claims associated with each, and the strong proofs presented as results. We kindly request the reviewers to thoroughly review these points, as they should provide a clearer understanding of the contributions and significance of our work.
>
> **Our Objectives (as presented in the paper):**
>
> > **1. Establishing a multi-stage temporal instillation objective:**
>
> The first stage focuses on learning to distinguish multiple sounds, while the subsequent stage emphasizes learning the temporal arrangement of these sounds.
>    - **Claim:** Training a single model to directly learn temporal relationships is ineffective because models must first develop the ability to distinguish between different sounds. Without this foundational understanding, learning temporality becomes infeasible.
>    - **Proof 1:** To validate this claim, we compared the model's performance after applying only TeminAL B with the performance after employing the multi-stage training process (TeminAL A followed by TeminAL B). The results show that the multi-stage model significantly outperforms in all tasks defined by ZSTE—a more advanced progression of the classical zero-shot evaluation strategy widely used in prior work. The results are detailed in Table 2.
>    - **Proof 2:** We further compared our model with state-of-the-art (SOTA) contrastively trained ALMs across all ZS evaluation tasks. Specifically, in ZSTE Task 2 (subtasks A and C), which require distinguishing multiple sounds, our model outperforms models trained on large datasets. However, in subtasks 2B and 2D (which involve simpler mappings of individual sounds), models trained on massive datasets excel due to their extensive exposure. For Tasks 3A and 3B, all models perform comparably, highlighting the challenges of zero-shot evaluations when trained on similar examples. These findings underscore the necessity of multi-stage training for better generalization.
>    - **Note:** Our training was intentionally restricted to a subset of the total parameters and used only the ESC-50 dataset. This decision aligns with our goal of demonstrating results within a realistic compute budget. Reviewers are kindly requested to consider that the focus of this work is not on achieving SOTA performance across billion-sample datasets, but rather on addressing a critical gap in current ALM post-training methods. Our key contributions lie in introducing a novel multi-stage training strategy and an effective evaluation framework.
>
> > **2. Demonstrating correct temporal relationships between audio and text:**
>    - **Claim:** Our analysis indicates that current contrastive ALMs face challenges in accurately capturing temporal relationships between audio and text.
>    - **Proof:** This claim is best evaluated after understanding our first objective. We acknowledge that the presentation order of objectives in the manuscript might have caused confusion, and we apologize for any inconvenience caused. However, the current order was chosen to maintain the logical progression of ideas.
>
> > **3. Proposing a general-purpose zero-shot evaluation scheme:**
>    - **Claim:** We are not claiming anything here, but providing and using a structured way of evaluating our earlier claims.
>    - **Proof:** Existing evaluation metrics fall short of effectively assessing temporal relationships, particularly in our case. We propose this evaluation scheme as a simple yet intuitive method to capture model behavior accurately. While future enhancements to evaluation metrics are certainly necessary, this scheme is sufficiently robust for our purposes.
>
> **Final Note:**
>
> If the addressed weaknesses and clarifications satisfactorily resolve the reviewers' concerns, **we kindly request an updated score** to reflect the potential impact of this work in advancing the field of audio-language models. Thank you.

---

> > ### Comment · Reviewer_isZu · 2024-11-25
> >
> > Thank the authors for addressing my concerns. I have increased my score.

---

> > > ### Author Response · Authors · 2024-11-25
> > >
> > > Respected Reviewer,
> > >
> > > We sincerely thank the reviewer for their thoughtful feedback and guidance, which have greatly contributed to improving our work. We appreciate your acknowledgment of the updates made in response to your earlier comments. We kindly request additional feedbacks on how we can further refine our manuscript to address your concerns and ensure it aligns with the standards for acceptance. Based on the current score, we feel that we may not have effectively communicated the key contributions and strengths of our work.
> > >
> > > Furthermore, **if our efforts to address your concerns regarding novelty, the proofs provided, and the discussions in the paper have met your expectations** and if the reviewer thinks the contribution in this work is novel and should be highlighted, then it's a humble request to score this work towards acceptabiity and help us share this knowledge towards the general development of the scientific community.
> > >
> > >
> > > With gratitude,
> > >
> > > --The Authors

---

> ### Author Response · Authors · 2024-11-28
> **Final Update**
>
> Respected reviewer isZu,
>
> Along with the above updates on your questions, we have taken the time to adress all the reviews and provide an in-depth detail of the summary (The response can be found up top in the comment section as **"Final updates and addressing the reviewers"** , https://openreview.net/forum?id=nplYdpc1Pm&noteId=tGw3QZiByT, Detailed paper review: https://openreview.net/forum?id=nplYdpc1Pm&noteId=lfCYvH8OhT) along with the limitations (https://openreview.net/forum?id=nplYdpc1Pm&noteId=jj9j5D2lq6) of our work. Kindly review this and if it satisfactorily answers your concerns then reward us with a better score.
>
> Thanks.
> --Authors.

---

> > ### Author Response · Authors · 2024-12-03
> > **Final day for reviews (2nd Dec)**
> >
> > Respected Reviewer isZu,
> >
> > We have carefully addressed all the concerns mentioned by the reviewers for improving our work, particularly regarding the limitations of our work. We humbly request you to kindly review the updates provided above in the detailed comment with links.
> >
> > We greatly appreciate your time and effort and respectfully request your consideration for a more favorable score.
> >
> > Thank you.

---

> > > ### Author Response · Authors · 2024-12-03
> > > **Buffer day request (3rd Dec, within the score update period)**
> > >
> > > Respected Reviewer,
> > >
> > > I hope this message finds you well. We have made a sincere effort to address all the concerns raised in your feedback to the best of our ability, ensuring that all limitations are clearly articulated. We kindly request you to consider updating the score in light of these revisions.
> > >
> > > Having worked meticulously on the revisions for over a month, we would greatly appreciate it if you could spare some time today to review them, as the opportunity for score updates remains available. Your thoughtful consideration would mean a great deal to us.
> > >
> > > Thank you for your time and understanding.
> > >
> > > Best regards,
> > >
> > > --Authors

---

### Official Review · Reviewer_YCxA · 2024-11-04

**Soundness:** 3
**Presentation:** 2
**Contribution:** 2
**Rating:** 5
**Confidence:** 5

**Summary:**

This paper proposes a post-training method to enhance CLAP's (Contrastive Language-Audio Pretraining) understanding of temporal relations in audio samples. This post-training method consists of two stages: (1) post-training on single and dual audio events, and (2) post-training on varying temporal sequences of the same event combinations. The paper also presents a detailed evaluation on zero-shot temporal evaluation (ZSTE) to demonstrate that CLAP, with the proposed two-stage training, achieves a better understanding of audio semantics in the temporal relations, especially on the understanding of both events in the overlapping cases.

**Strengths:**

The strengths of this paper lie in two main areas:

(1) **Two-Stage Post-Training**: The paper introduces a two-stage post-training method based on an assumption about the learning challenge in CLAP models, which is the difficulty gap related to the number of events versus their temporal order. By addressing these learning challenges progressively through distinct post-training stages, the paper demonstrates performance improvements. Additionally, it contributes by adapting the training objectives of CLAP to TNCE loss terms, though this is primarily referred by prior work in computer vision.

(2) **Zero-Shot Temporal Evaluation**: The introduction of ZSTE establishes a benchmark for audio understanding. The paper provides a range of tasks to thoroughly evaluate CLAP's performance in understanding temporal relations in audio samples.

**Weaknesses:**

However, the weaknesses of this paper are substantial and can be categorized into three main points:

(1) **Limited Novelty and Generalization**:  The primary contribution lies in the two-stage post-training approach and its experiments; however, the concept of using augmented captions (such as concatenation and overlay) to improve CLAP’s understanding of temporal audio relations has already been explored in prior works [1][2]. Moreover, the paper lacks sufficient evidence and explanation to justify the necessity of a two-stage post-training approach (see point (2) below). Generalization is also a critical concern, as the paper primarily focuses on two-event audio scenarios and restricts prompts to a narrow set of captions—[before, after, while]. Real-world audio scenarios are significantly more complex, involving a wider variety of events and more natural language descriptors. Scaling this method to accommodate more events and captions poses a significant challenge, as the size of the contrastive matrices would become prohibitively large for processing.

(2) **Inadequate Experimental Setup to Demonstrate Effectiveness**:

a. While the proposed post-training method could be applied to existing CLAP models (e.g., LAION-CLAP, Microsoft-CLAP, CompA-CLAP), the authors did not showcase this flexibility. As mentioned in Section 3.4, “the encoders need to be pretrained", it would have been more effective to apply the post-training method to 2-3 different CLAP models and compare their performance with and without the post-training. This approach would demonstrate the post-training method's effectiveness without the need to train the CLAP model from scratch (as it appears the authors did) to save time. It also strengthens the evidence supporting the proposed method.

b. Effectiveness of Two-Stage Training (i.e., TerminAL A and B): Table 2 presents a partial ablation study but omits the results for TerminAL A-only. Including this result would provide a clearer picture of how TerminAL A alone impacts ZSTE performance, offering more comprehensive evidence for the individual effectiveness of each training stage.

c. Overall Performance: According to Table 3, the proposed T-CLAP does not consistently outperform previous models or yield comparable results on non-temporal tasks. CompA-CLAP and T-CLAP each achieve the top-1 performance on approximately half of the tasks. While CompA-CLAP struggles with tasks 2A and 2C, which require predicting both events, T-CLAP sacrifices substantial accuracy on task 1A (dropping from 82% to 75%), which is the most fundamental audio understanding task. These results suggest two conclusions: first, the single-dual contrastive training (T-A) step is crucial, as it drives improvements in tasks 2A and 2C. Second, a trade-off remains between temporal enhancement and core audio classification accuracy, as the proposed method fails to preserve the performance on basic classification task, even falling below ML-ACT, (the baseline before CLAP).

(3) **Poor Presentation**:  This paper lacks several necessary introductions on its regular pages. The paper should emphasize more on explaining the T-A and T-B training paradigms (see further details in the question section) instead of reiterating TNCE details, which are largely sourced from [3]. Additionally, the experimental section is hard to follow due to minimal guidance on the five tasks and their A-B(-C-D) variations. The analysis of these experiments is too brief, making it difficult to validate the effectiveness of the two-stage training method and assess the overall performance. Furthermore, the title of the paper is vague and somewhat misleading; it would benefit from a clearer focus, such as emphasizing **temporal relation enhancement** in **contrastive language-audio pretraining** models.

Above all, I found this paper falls below the standard of ICLR, for its limited novelty and generalizability to enhance CLAP performance; insufficient experimental setup to effectively validate the method’s impact and stability in audio understanding through post-training; and a lack of writing organization that hinders clear communication of the essential training paradigms.

[1] CompA: Addressing the Gap in Compositional Reasoning in Audio-Language Models

[2] T-CLAP: Temporal-Enhanced Contrastive Language-Audio Pretraining

[3] Test of Time: Instilling Video-Language Models with a Sense of Time

**Questions:**

1. In the TerminAL A training stage, the model is fed both single-event and dual-event samples. Do these samples also include overlay cases? For instance, will "Dog" and "Dog while Cat" be presented in the same batch?

2. In Table 2, T-AB appears to improve the 2A and 2C tasks compared to T-B. This suggests that T-A introduces a bias that enhances performance. However, I am unclear on how the T-A training stage leads to this bias. If "Dog" and "Dog while Cat" are presented in the same batch, the samples for "Dog while Cat" will still be penalized under the "Dog" label. Could you provide more insight into how the model effectively retrieves single-class captions when presented with dual-event (overlay) audio samples?

3. Were additional temporal-enhanced captions considered for training, or was the model exclusively trained with prompts like [after, before, while]? For instance, were alternatives such as [then, followed by, and then] also utilized?

4. The use of prompts when combining two audio events in CompA-CLAP [1] differs from the proposed TerminAL approach. Do you believe this discrepancy will affect the performance results presented in Table 3 and whether the comparison remains fair? For example, it is possible that the keywords [before, after] were not employed in the training of CompA-CLAP.

[1] CompA: Addressing the Gap in Compositional Reasoning in Audio-Language Models

---

> ### Author Response · Authors · 2024-11-16
>
> We are thankful for the insightful reviews and comments on our work. We are working on refining our subsmission and adressing all the questions marked in the review. Looking forward towards this discussion.
>
> Thanks,
> --Authors.

---

> ### Author Response · Authors · 2024-11-21
>
> We sincerely thank the reviewers for their thoughtful feedback. Below, we provide a detailed response to each identified weakness, address the specific questions raised, and outline the changes **incorporated into the revised submission**.
>
> We respectfully request the reviewers to carefully consider our explanations and arguments, as we believe they are integral to advancing the understanding we aim to contribute to the community through this work.
>
> **Weaknesses:**
>
> > **W1. Limited Novelty and Generalization**
>
> We thank the author for mentioning the two--stage training contribution for our work. And we do agree that the method used in our work has been explored in the past. However we don't think one can look at these two aspects individually, being able to instill time through contrastive training on modified samples is a general way of doin this, but doing it correctly is what we have emphasised in our work. And in order to provide evidence to our claims we have presented sufficient results in section 5. Although we humbly accept that earlier the results were not presented well in the document. However, we have made necessary changes in order to get across our message. The following is our take on how we prove our claims around novelty and generalisation.
>
>    - **Claim:** Training a single model to directly learn temporal relationships is ineffective because models must first develop the ability to distinguish between different sounds. Without this foundational understanding, learning temporality becomes infeasible.
>    - **Proof 1:** To validate this claim, we compared the model's performance after applying only TeminAL B with the performance after employing the multi-stage training process (TeminAL A followed by TeminAL B). The results show that the multi-stage model significantly outperforms in all tasks defined by ZSTE—a more advanced progression of the classical zero-shot evaluation strategy widely used in prior work. The results are detailed in Table 2.
>    - **Proof 2:** We further compared our model with state-of-the-art (SOTA) contrastively trained ALMs across all ZS evaluation tasks. Specifically, in ZSTE Task 2 (subtasks A and C), which require distinguishing multiple sounds, our model outperforms models trained on large datasets. However, in subtasks 2B and 2D (which involve simpler mappings of individual sounds), models trained on massive datasets excel due to their extensive exposure. For Tasks 3A and 3B, all models perform comparably, highlighting the challenges of zero-shot evaluations when trained on similar examples. These findings underscore the necessity of multi-stage training for better generalization.
>    - **Note:** Our training was intentionally restricted to a subset of the total parameters and used only the ESC-50 dataset. This decision aligns with our goal of demonstrating results within a realistic compute budget. Reviewers are kindly requested to consider that the focus of this work is not on achieving SOTA performance across billion-sample datasets, but rather on addressing a critical gap in current ALM post-training methods. Our key contributions lie in introducing a novel multi-stage training strategy and an effective evaluation framework.
>
> > **2. Demonstrating correct temporal relationships between audio and text:**
>    - **Claim:** Our analysis indicates that current contrastive ALMs face challenges in accurately capturing temporal relationships between audio and text.
>    - **Proof:** This claim is best evaluated after understanding our first objective. We acknowledge that the presentation order of objectives in the manuscript might have caused confusion, and we apologize for any inconvenience caused. However, the current order was chosen to maintain the logical progression of ideas.
>
> > **3. Proposing a general-purpose zero-shot evaluation scheme:**
>    - **Claim:** We are not claiming anything here, but providing and using a structured way of evaluating our earlier claims.
>    - **Proof:** Existing evaluation metrics fall short of effectively assessing temporal relationships, particularly in our case. We propose this evaluation scheme as a simple yet intuitive method to capture model behavior accurately. While future enhancements to evaluation metrics are certainly necessary, this scheme is sufficiently robust for our purposes.
>
> In order to make the submission better and to encompass everything which we state above, the whole Section 4.2 has been revised to incorporate necessary information from Appendix Section B.4. Due to space limitations, we have moved some background details and additional diagrams to the supplementary section. Nonetheless, all important impacts and applications of ZSTE are now detailed in the Results section (Section 5) as we have re-written the whole section. The results section now includes Table 1, Table 2, and explanatory paragraphs in Section 5.

---

> > ### Author Response · Authors · 2024-11-21
> >
> > **Weaknesses:**
> >
> > > **W2. Inadequate Experimental Setup to Demonstrate Effectiveness**
> >
> > - Generalising on different existing CLAP models: The reviewer is correct to note that we can extend this to other methods, but following the objectives of our current work this kind of generalisation is not just out of scope but also out incorrect. (i)  Firstly, we are proposing a scheme which tries to instill the sense of time in an existing ALM, henceforth performing this on a model which has already tried to align its weight in a different direction would not necessarily be the best thing to do. (One can think this in terms of fine-tuning an already fine-tuned model, which more often than not entails hallucinations. As it will bring in new information over the existing information, [1] Gekhman et. al.). (ii) Secondly, our objective is to propose a method which can correctly align a model, henceforth comparison with other model on similar task is the sufficient condition which is required for objective. We never comit to a proposing a generalised model which will be competent on any task (**And any ML model which claims to do that, is somewhere or the other limiting its scope of generalisability without asserting it explicitly.**)
> >
> > Furthermore, our approach achieves significant advancements within a limited computational budget, training around 10% of the total trainable parameters and utilizing a single dataset (ESC-50). Unlike prior works, such as those by Ghosh et al. (2023) and Wu et al. (2023a), which rely on more expansive datasets and substantial computational resources. Our focus is on developing a methodology that can effectively instill a sense of time in the model within acceptable computational constraints, rather than on generalizing over large, diverse dataset.
> >
> > > **W2. Poor Presentation :**
> >
> > - We humbly accept the reviewers comment on our presentation, although suggestions on this would have been helpful. As of now the weakness mentioned appears vague. However we take this comment seriously and have re-written the whole section 4 and 5 along with major changes in section 1 (lines 93-106) to better convey the results and the sufficient conditions to our claims and objectives.
> > - The following are the details on the sections which the reviewer asks for:
> > - **T-B and T-AB (T-A and T-B):** Actually T-B : only TeminAL B trained model and T-AB : TeminAL A followed by TeminAL B , we explain the details on the different objectives in Section 3.4 : The different objective function for TeminAL A and TeminAL B , Appendix B.3: continuation of objective function derivation (in supplementary as limited space in the main) , Figure 3 : Schematic of the overall training of TeminAL B , Figure 4 : Following Figure 3, we expand the training objective schematically and Figure 7, we show similar training objective schematic for TeminAL A.
> > - **ZSTE tasks (5 tasks ):** We accept the unclear descriotion of this section and hence we changed section 4.2 and 5 to better convey the impact of these tasks and the results of various models. Morever we have a detailed section descibing each of these tasks in section B.4. We request the reviewer to make use of the supplementary section for clarity, we couldn't fit this in the main section due to page restrcition. If required we can submit this as a separate supplemnetary document in our submission.
> > - **Title of the paper:** We are thankful to the reviewer for pointing out the lack of specificity in the title (essentailly benefit of participating in a peer review process),  and we humbly accept the marked improvement, we have made the title more specific **Temporal Enhancement of Contrastive Audio--Language Models through Self--Supervised Post--Training with Text-Audio Pairs** . We look forward to more suggetion on this in order to make a better submission overall.
> >
> > [1] https://arxiv.org/abs/2405.05904

---

> > > ### Author Response · Authors · 2024-11-21
> > >
> > > **Questions:**
> > >
> > > > **Q1. In the TerminAL A training stage, the model is fed both single-event and dual-event samples. Do these samples also include overlay cases? For instance, will "Dog" and "Dog while Cat" be presented in the same batch?**
> > >
> > > - No, the training stage TeminAL A does not contain overlaying samples. We have made this on [246-249], Figure 7 and Figure 6 , we have now included both these figures for better clarity on the objective and the batch of data. We thank the reviewer for this question and hope we are now able to put this better.
> > >
> > > > **Q2. Question on the impact of TeminAL A training stage**
> > >
> > > - That's excellent, we thank the author to note the results from Table 2 to correctly infer the impact of our strategy on the performance. We think the confusion occured due to not clear understanding of Q1, so following Q1. We don't train on overlapping samples, rather in stage 1 (TeminAL A) we only want the model to differentiate 2 sounds, so we train with single source sound and a corrosponding concatenated sound. Essentially an untrained model doesn't know how to differentiate the two sounds and hence will give equal probable similarity to score to both, and that's where we wish to penalise the model thorugh our loss function and hope that a well optmised training routine can then capture these penalty and align the model towards distinguishing multiple sounds.
> > >
> > > - And essentailly that's what you observe in 2A and 2C, because these are the tasks which exlicitly want the model to classify both audio distinctly. While other models are well performing on 2B and 2D and 3A and 3B, where we only ask them to pick at-least one class and map to a single temporal label respectively, they are not able to perform well on 2A and 2C.
> > >
> > > - Moreover, better results on 3A and 3B shows that a model which has been trained on an extensive dataset, may well perform well on similar temporal samples as seen from it's training routine even if it's not able to udnerstand multiple sounds separatly (Essentially 2A and 2C). We hope this answer satisfies raised query.
> > >
> > > > **Q3. Temporal samples for training**
> > >
> > > - No, we only train on these set temporal behaviour, as mentioned earlier the objective of the work is not on achieving SOTA performance across billion-sample datasets or across all possible temporal generalisation, but rather on addressing a critical gap in current ALM post-training methods. Our key contributions lie in introducing a novel multi-stage training strategy and an effective evaluation framework.
> > >
> > > - The question can again be mapped to covering generalisabilty, and we would like to reiterate the answer from W2.
> > >
> > > > **Q4. CompA--CLAP training disparity with our evaluation**
> > >
> > > - One of the important and excellent approach used in CompA is their template creation module, which uses an open-ended language model (GPT in their case) to generate general positive and negative samples. As of now it will be difficult if they did not have "before" and "after" like prompt in their dataset or not. That's essentailly why we have our strategised evalaution metric, the performance of the model on Task 3 essentially conveys hat the model is well capable (and even better than our model) in detecting the temporal event with these prompts.
> > > - Although generalising to that extent either in terms of dataset size or parameters is ou of scope for our current budget, and hence we have tried to contruct this work from initiation around a defined set of objectives respecting our compute limitation.
> > >
> > > **Final Note:**
> > >
> > > If the addressed weaknesses and questions satisfactorily resolve the reviewers' concerns, **we kindly request an updated score** to reflect the potential impact of this work in advancing the field of audio-language models. Thank you.

---

> ### Comment · Reviewer_YCxA · 2024-11-24
> **Feedback on the Authors Response**
>
> I really appreciate the response from the authors on my concerns on this submission.
> I carefully read all these responses and list my feedback below:
>
> I agree the below responses and will update the score accordingly:
>
> 1. The revision on section 4.2, the change of the title, and the revision on demonstrating the TB / T-AB experiments and 5 ZSTE tasks make the presentation of this paper more clear.
>
> 2. The contribution of proposing a ZSTE benchmark is valuable.
>
> I disagree the below responses and give my reasons on it:
>
> 1. The underperformance from the proposed T-CLAP to the previous models (e.g., CompA) on subtasks 2B and 2D is due to to the limitation of the dataset and computational resources.
>
> Concern: **there are some big changes on the paper statements and motivations for the revised version.** Unfortunately Openreview does not allow to access the paper at the version before rebutall (Nov 19). But I do have one saved copy when I reviewed the paper:
>
> 1.a. In the revised submission and the rebuttal, "**In contrast, our approach achieves significant advancements within a limited computational budget,
> training around 10% of the total trainable parameters and utilizing a single dataset (ESC-50)**."
>
> 1.b. In the original submission, the above statement **was not presented**.
>
> The authors explained the reason of underperformance of subtasks 2B and 2D on the limited datasets and computational resources and this becomes one of their proposed contribution. I strongly disagree with this point not only because this was not stated in the original submission, but also given the fact that:
>
> 1.c. For a contrastive learning model that commonly requires a large-scale training paradigm, it remains very unconfident that if the proposed results actually overfit to the ESC-50. I would like to request the result of ZSTE on other testset (such as processed one from DESED, UrbanSound, AudioSet Strong, etc.)
>
> 1.d. Admittedly, the pre-training of the CLAP model involves more datasets listed in Appendix B.5-Table 6. In that, we actually cannot conclude that the proposed method is a computational effective method "just because the post-training process only involves ESC-50". In fact, this makes me confuse about another statement the authors made in section 4.1: "**We train a CLAP model Elizalde et al. (2023) using transformer-based encoders: HTSAT Chen et al. (2022) for audio and BERT Devlin et al. (2018) for text, each with a projection layer.**" So it seems like the authors never trained the CLAP model from scratch but just used some pre-trained checkpoints from the previous works (and then did the post-training with ESC-50?
>
> In summary, I still cannot fully assess the effectiveness of the proposed "correct method on temporal enhancement". This still aligns my initial statement: "**it would have been more effective to apply the post-training method to 2-3 different CLAP models and compare their performance with and without the post-training.**"
>
> Minor comments:
>
> 2. It is still valuable and necessary to have Teminal-A-only ablation study on the results to fully **assess the effectiveness of Terminal-B**. Otherwise it would be really uncertain how Terminal-B method can further improve the temporal awareness of the model reflected in the performance.
>
> 3. The generalization of the temporal template words (before, after, while) should be further considered to add more words.
>
> In summary, I raise my score as **5** for the revision of the paper representation and organization, but remain unsatisfied with other aspects of the submission.

---

> ### Author Response · Authors · 2024-11-25
>
> We sincerely appreciate the reviewer for taking the time to provide highly valuable comments on our responses. To address the concerns efficiently, we wish to provide our replies to the above-mentioned comments. If necessary, we will offer an extended response before the final interaction window to ensure that our message is conveyed completely and accurately.
>
> > **1. Responding to the weakness mentioned by the reviewer :**
>
> - **1a. and 1b. Changes in the motivation:** We accept that we have made changes in the motivation section, but we have not tried to re-align the paper. Although we do accept that the initial message was not coming though well, and all the credits for the update only goes to the insightful reviewers of the paper. But I wish to present examples from the original submitted paper, where we have explicitly mentioned the limited budget post--training of CLAP on a limited dataset (here being ESC--50) , <The following are lines and sections from the original submission on 10/01/2024>
>
> -  > a) Figure 2 and 3 : Here we had schematically shown only a subset of the layers as trainable, while adapting the origianl encoders with temporality.
> - > b) Line 310 : Encoders are not trained from scratch; we extend our framework using a pre- existing audio-language model comprising an audio encoder $fc_\theta$ and text encoder $fc_\phi$ as shown in figure 3.
> - > c) Line 210 : Details of our study on the limited dataset ESC--50. Followed by section B2 for the details.
> - > d) Line 1271 : The whole section B.5 of the original submitted paper mentions our training paramters as well as cost of training.
>
> - **Accepting sections from our original submission which were confusing:** We would also like to humbly accept the confusing section of the initial submission which may have resulted in the confusion of the intent. Again all thanks to the reviewers for their continued effort and insightful suggestions.
>
> - > a)  Line 314 : Due to limited dataset size, selective refinement of specific layers within $\Theta$ is performed. Here we wanted to convey the message of trade-off between dataset size and trainable parameters in the perspective of overfitting and budget, but failed to do so in the original submission.
> - > b) Section 4.1 : We accept the confusing lines in this section, we wanted to convey what we train on the baseline model. We are very thakful for the the reviewer for pointing out this confusion. We have updated our section in the main paper to avoid this confusion.
>
> - **1c. : Results overfitting on ESC--50:**
>
> - We do accept this remark from the reviewer and think it's an important remark in general toward model post--training. Although, as we emphasis in our objective and scope of work, that our aim is not give a generalisable model which perform SOTA across all datasets. Rather our aim is to present the idea of multi--stage training of contrastive models and it should effectively be done in the sense of instillation of time.
>
> - > **Hypothesis for 1c. :** Evaluating results of other mentioned dataset would not be able to give better result than the origianl CLAP model, as we never post--trained on those specific datasets. Henceforth encountering any OOD sample would mean feeding unkown data to the model to hallucinate on. While the training can well be extended to different datasets or to a better mixture of datasets, but that defeats the purpose of being able to train within our compute budget. Although this point has been well noted for future work which will emphasis solely on giving generalisable post--trained models. **We request the reviewer to share their view on our hypothesis**.
>
> - **1d. Confusing statement on whether we have fully trained our own CLAP model or not?**
>
> We again thank the reviewer for pointing to this confusion. The reviewer is absolutely correct in noting that we never train the CLAP model from scratch, we rather align the pre-trained model with our post--training strategy. As mentioned in the response of 1.a. we humbly accept these confusing lines and have updated the section 4.1 for better clarity.
>
> **Note :** We would also wish to make a remark on authors comment on our method being **computational effective method**, we will abstain ourselved for making that statement, as compute effective and compute efficient training are broader scope terms which would require heavy abalations for prooving. However, we would wish to only inlcude statements on compute limitation and budget, which does distinguish our work from others but only in terms of providing an effective multi--stage training strategy rather than giving a compute efficient generalisable model which is performant across all data distributions.

---

> ### Author Response · Authors · 2024-11-25
>
> We highly value the reviewer's summary, although we would humbly wish to share our response such that our objectives and scope are communicated well.
>
> > **Response to reviewer's summary**
> - We accept the remarks by the reviewer on viewing the effectiveness through post--training on multiple different datasets. We would like to present our thoughts both positive and counter-positive (not in the theoretical sense, but in the terms of scope of our current work.)
>
> - > **Increasing knowledge base of the model:**  The positives is fairly straightforward, as we would get better underdantind on how the method would perform on variations of differnet kinds of datasets. However, we would like to comment that since the data space of current Audio--Language contrastive models are all based on single event monoaural sounds, with non-descriptive caption. The extension of the dataset would only increase the knowledge base of the current model and hence--forth better performance. Although it would not bring a fundamental change in terms of generalisable multi--modal learning. And as stated previously our work focuses only on providing an effective multi--stage training strategy rather than giving a compute efficient generalisable model which is performant across all data distributions.
> - > **Inceraisng model complexity to overcome over-fitting** :  Extension on multiple different datasets would effectively involve incereasing the learnable parameters of the model to avoid over-fiting of the models which makes the work go out of scope in terms of compute. Hence, defeats our objective of the project.
>
> > **Response to minor comments**
>
> > **2. Providing abalation for TeminAL A:** We note the reviewer's concern on providing abalation study for TeminAL A. However, we would like to put forth our thinking about judging effectiveness through providing abalation on TeminAL B and TeminAL A+B. We would humbly like to suggest that, the abalation presented in Table 2 and results of Table 3 between TeminAL B and TeminAL A+B, actually serves this purpose and no new information can be broghut upon by abalation on TeminAL A.
> - > It may help by thinking this way, we wish to capture the gain due to TeminAL A, hence model with TeminAL A (A+B) and model without TeminAL A (Only B), are the necessary and sufficent requirements. Assume model tarining as vectors getting scaled and transformed, initial vector be $v_i$.
> - > Let TeminAL A transform it to $v_a$ (scale by $\alpha$ and rotate by $\theta$) and further teminal B transform it to $v_{ab}$ (further scale by $\beta$ and rotate by $\phi$), while direct transformation would be $v_b$ (i.e $v_i$ scale by $\beta$ and rotate by $\phi$) . Henceforth $v_b$ is actually $v_i$ scaled by $\beta * \alpha$ and rotate by $\phi + \theta$. Hence, even without the information of the transformation incurred due to TeminAL A (i.e $v_a : $ $\alpha$ and $\theta$), we can infer it through only $v_b$ and $v_{ab}$. Application of monotonic non-linearity will make the above non-computational and hence we won't comment on quantitative aspects (i.e how much effect, or exact values of $v_a : $ $\alpha$ and $\theta$), but qualitatively the general trend would preserve.
>
> > **3. Addition of temporal words :**  We accept the suggestion of the author that increasing the context of temporal words would increase the generalisabilty. But would humbly argue that this would still not provide a generalisable solution (something which we are currently working on as well). Again circling back to the scope of the current work, that we neither wish nor are capable enoguh in this project to provide a generalisable model. Although we only focus on providing an interpretible way of multi--stage contrative training of ALMs to instill the specific case of temporality.
>
> **Final remarks and request :** We sincerely appreciate the reviewer for taking the time to provide highly valuable comments on our responses. We highly value the challenges mentioned by the reviewer but we also like to mention that it would be out of our scope to tackle all the challenges in one work. **However, If the addressed weaknesses and clarifications satisfactorily resolve the reviewers' concerns specifically to our current scope, then we kindly request the reviewer to reward us a better score.**
>
> Thank you.

---

> ### Author Response · Authors · 2024-11-28
> **Final update**
>
> Respected reviewer YCxA,
>
> Along with the above updates on your questions, we have taken the time to adress all the reviews and provide an in-depth detail of the summary (The response can be found up top in the comment section as **"Final updates and addressing the reviewers"** , https://openreview.net/forum?id=nplYdpc1Pm&noteId=tGw3QZiByT, Detailed paper review: https://openreview.net/forum?id=nplYdpc1Pm&noteId=lfCYvH8OhT) along with the limitations (https://openreview.net/forum?id=nplYdpc1Pm&noteId=jj9j5D2lq6) of our work. Kindly review this and if it satisfactorily answers your concerns then reward us with a better score.
>
> Thanks.
> --Authors.

---

> > ### Author Response · Authors · 2024-12-03
> > **Final day for reviews (2nd Dec)**
> >
> > Respected Reviewer YCxA,
> >
> > We have carefully addressed all the concerns mentioned by the reviewers for improving our work, along with including the limitations of our work. We humbly request you to kindly review the updates provided above in the detailed comment with links.
> >
> > We greatly appreciate your time and effort and respectfully request your consideration for a more favorable score.
> >
> > Thank you.

---

> > > ### Author Response · Authors · 2024-12-03
> > > **Buffer day request (3rd Dec, within the score update period)**
> > >
> > > Respected Reviewer,
> > >
> > > I hope this message finds you well. We have made a sincere effort to address all the concerns raised in your feedback to the best of our ability, ensuring that all limitations are clearly articulated. We kindly request you to consider updating the score in light of these revisions.
> > >
> > > Having worked meticulously on the revisions for over a month, we would greatly appreciate it if you could spare some time today to review them, as the opportunity for score updates remains available. Your thoughtful consideration would mean a great deal to us.
> > >
> > > Thank you for your time and understanding.
> > >
> > > Best regards,
> > >
> > > --Authors

---

### Author Response · Authors · 2024-11-22
**Follow-Up on Rebuttal Responses and Score updates.**

Dear Reviewers,

Thank you for your valuable feedback. We greatly appreciate the insights you provided, which have helped us improve our submission.

As the author interaction window approaches its conclusion **(26th Nov)**, we kindly ask you to review our rebuttal responses and consider whether they adequately address your concerns. If you find that all your concerns have been addressed, **we would humbly request you to consider updating your scores accordingly.**

Thank you.

---

> ### Author Response · Authors · 2024-11-23
> **2nd Notification**
>
> Dear Reviewers and Chairs,
>
> We would like to ask the Chairs to kindly notify the reviewers as well. As the author interaction window approaches its conclusion **(November 26 at 11:59 pm AoE)**, we kindly ask you to review our rebuttal responses and consider whether they adequately address your concerns. If you find that all your concerns have been addressed, we would humbly request you to consider updating your scores accordingly.
>
> Thank you.

---

> > ### Author Response · Authors · 2024-11-24
> > **3rd Notification**
> >
> > Dear Chairs,
> >
> > We would like to ask the Chairs to kindly notify the reviewers as well. As the author interaction window approaches its conclusion **(November 26 at 11:59 pm AoE)**, we kindly ask them to review our rebuttal responses and consider whether they adequately address their concerns.
> >
> > Thank you.

---

> > > ### Comment · Area_Chair_LtVd · 2024-11-24
> > >
> > > Yes, reviewers have been reminded to respond.
> > >
> > > Thanks,
> > > AC

---

> ### Author Response · Authors · 2024-11-26
>
> Respected Chair,
>
> Thanks a lot for you help. Following the author's guide (https://iclr.cc/Conferences/2025/AuthorGuide), may you kindly tell us how to update our paper's title, we are unable to do so with the current frontend setup. We wish to change our paper's title to **"Temporal Enhancement of Contrastive Audio--Language Models through Self--Supervised Post--Training with Text--Audio Pairs"**. This has already been communicated to the reviewers (https://openreview.net/forum?id=nplYdpc1Pm&noteId=04VFUnDzfs). Moreover, this is accounting the suggestion given by the insightful suggestion of Reviewer YCxA.
>
> Sincerely,
>
> --Authors.

---

> > ### Author Response · Authors · 2024-11-29
> > **Notification to the reviewers**
> >
> > Dear Reviewers,
> >
> > Thank you for your valuable feedback. We greatly appreciate the insights you provided, which have helped us improve our submission.
> >
> > As the author interaction window approaches its conclusion **(2nd Dec)**, we kindly ask you to review our rebuttal responses and the final draft with all the asked changes. If you find that all the concerns have been addressed, **we would humbly request you to consider updating your scores accordingly.**
> >
> > Thank you

---

> > > ### Author Response · Authors · 2024-11-30
> > > **2nd notification to the Reviewers. (T-2 days)**
> > >
> > > Dear Reviewers,
> > >
> > > Thank you for your valuable feedback. We greatly appreciate the insights you provided, which have helped us improve our submission.
> > >
> > > As the author interaction window approaches its conclusion in 2 days **(2nd Dec)**, we kindly ask you to review our rebuttal responses and the final draft with all the asked changes. If you find that all the concerns have been addressed, we would humbly request you to consider updating your scores accordingly.
> > >
> > > Thank you

---

### Author Response · Authors · 2024-11-28
**Final updates and addressing the reviewers**

The following is our final summary on the paper, the highlight of final revisions are:

- **Requsted changes:** All the necessary changes which are highlighted by our reviewers in order to make the paper impactfull (Throughout the paper).
- **Major contributions:** (Line 129), such that we don't include something which we haven't proved in the paper.
- **Clarity on training:** Worked on explicit clarity on our 2-stage training scheme and objective function (inlcuded all relevant things for TeminAL A in the paper Figure 6,7 and Section B3)
- **Limitations of the model:** We also present the limitation of our model and address that inspite of having these limitations we never go out of the scope of our project (Section B7).
- **Clarity on fairness of evaluation:** We address the reviews on fairness of evalutation in section (Section 5 as well as majorly in the comments). The model is trained on the training set of ESC-50 likewise all the other models, while evaluations are performed on the evaluation split. Although the other models have seen vast amount of data (a lot of which have overlapping classes, check Audiosets, Clotho and ESC-50), so a model which may be at a disadvantage is ours. Although we acknowledge that our model also views these data during pre-training of CLAP (not done by us).
- **Type errors:** We have tried to correct all the type-errors and synactic errors in the paper and the figures.

---

> ### Author Response · Authors · 2024-11-28
> **Full summary of our work (every line included) after the rebuttal rounds.**
>
> - **Summary:**
>
> > 1. [Lines 38-49]: Multimodal learning and its capabilities towards progressing aritifical general intelligence. Here, we particularly focus on contrastive learning models and its applications, specifically focussing on Image-Language CLIP [1].
> > 2. [Line 50-86]: Extending contrastive models to different modalities like Audio-Language model (ALM) CLAP [2] and also highlighting issues in the current ALMs particularly having bias towards specific words rather than their arrangment or context [3]
> > 3. [Line 87-96]: We describe various previous model and how they adress the drawbacks stated above. Mainly focussing on Compa-CLAP [4] and how it addresses the problem with attribution and composition training.
> > 4. [Line 95-105]: We address the challenges which previous model face, specifically (a) Distinction of multiple sounds before instilling the understanding of temporality, (b) Extending upon this understanding towards building temporal understanding. Contrasting to the previous approaches we implement while operating in a limited computational budget while training only 10% of the total trainable parameter of the base CLAP model.
> > 5. [Line 106-119]: A brief overview on our model and training for the introduction. Details of which is discussed later in specific section
> > 6. [Line 120-127]: Our introduction to the ZSTE evaluation and why it's needed as cited be previous authors. We abstain from claiming it to be general purpose as we have not shown it to be. Although intuitively the scheme can be generalised to other modality with ease.
> > 7. [Line 128-144]: Our main contribution, claims and proofs of which have already been detailed in https://openreview.net/forum?id=nplYdpc1Pm&noteId=YpDrcK1rig for the readers.
> > 8. [Line 148-161]: Background on different audio-language models both open and closed ended and their applications. Emphasising on why the work is important and relevant for the multi-modal articifial intelligenece community.
> > 9. [Line 164-176]: After intoroducing the importanace of ALMs, we wish to intorduce previous applications of Self-supervised learning and Post-training particularly in the domain of contrastive learning, so that the reader gain context on the domain we wish to operate on.
> > 10. [Line 179-194]: Finally in the background section, we wish to cover Zero-shot models and its limitations, as we further propose our own scheme for zero-shot evaluation (ZSTE). Sumarising the limitations, classical ZS inference schemes may lack to properly capture the context (in whichever sense, be it attribution, POS or temporal).
> > 11. [Line 199-220]: In order to address our objectives which may be used to solve the problems with ALMs and Evaluation schemes which have been listed in the previous sections, we propose our method focussing on contrastive training of a pre-trained ALM (CLAP) with temporally augmented samples.
> > 12.[Line 296-313]: We introduce our objective function for training which contains a 2-step training objective TeminAL A which is used to distinguish multiple sounds  followed by TeminAL B used to contrastively train to instill the sense of time in it.
> > 13. [Line 314-315]: Information on how we form our batches of data following section 3.2 for the actual training, followed by introduction to our limited parameter training (limiting to specific few layers of the encoders.) with the net Loss function $L_B$= $L_{c_B}$ + $\beta$ $(L_{a_B})$
> > 14. [Line 331-377]: Defining the textual Loss $(L_{c_B})$ of Terminal B in eq(3) followed by defining all it’s terms in eq 4,5 and 6. Followed by Defining audio loss $(L_{a_B})$ in eq(7) followed by all its terms in eq 8 and 9. Similar formulation of loss function for Teminal A can be found in appendix B.3.
> > 15. [Line 407-422]: The impact of hyper-parameters $\(\{\alpha_s, \alpha_c, \alpha_{st}, \alpha_{ct}\}\)$ and $\(\{\beta_A, \beta_B\}\)$ on the temporal sensitivity of encoders. These coefficients enhance sensitivity to time-reversed samples and dataset adaptability, enabling the model to treat similar and dissimilar pairs differently. The inclusion of these hyper-parameters refines the encoder's temporal understanding and ensures distinct sample treatment.
> > 16. [Line 429-431]: Introduction to our base model CLAP and its components.
> > 17. [Line 434-443]: Our model evaluation method ZSTE and downstream tasks (Audio & Text retrieval and Audio Classification). ZSTE is divided into 5 tasks where Task 2,3,4 and 5 have 4,2,2,2 subtasks each. While downstream tasks are evaluated for T-A and A-T retrievals on AudioCap/ Clotho dataset. And classification is evaluated on (ESC-50 and US8K dataset).

---

> > ### Author Response · Authors · 2024-11-28
> >
> > > 17. [Line 434-443]: Our model evaluation method ZSTE and downstream tasks (Audio & Text retrieval and Audio Classification). ZSTE is divided into 5 tasks where Task 2,3,4 and 5 have 4,2,2,2 subtasks each. While downstream tasks are evaluated for T-A and A-T retrievals on AudioCap/ Clotho dataset. And classification is evaluated on (ESC-50 and US8K dataset).
> > > 18. [Line 459-473]:Result discussion on downstream tasks which were mentioned in 2.3, we observe that the trained T-CLAP model doesn’t forget. The results are shown in Table 1 and 4, showing that the model still remains competitive on the downstream tasks. Although the datasets used for evaluation do have overlapping sounds/ texts with our training dataset; but so is the same for all the models (shows a general drawback in evaluation datasets and weaknesses of classical metrics for evaluation/)
> > > 19. [Line 475-523]: Next up we discussed the results of hyper-parametric variations (or abalations) to determine the impact of our loss function modification. The results are shown in Table 2, which can be well summarised as the impact of our objective function modification as well as the 2-stage training procedure.
> > > 21. [Line 540-554]: Finally for the result section, we demonstrate how our model T-CLAP compares to the other state of the art contrastive models on the ZSTE evaluation metric. Note: Each of the mentioned models have used the ESC-50 dataset for their training along with use of ESC validation set is a fair choice. Comparing all the models we find that T-CLAP is better in terms of distinguishing overlapping sounds. While remaining competitive on other tasks as well.
> > > 22. [Line 558-571]: In conclusion, we demonstrate the need of a temporal instillation scheme of contrastive learning model which performs better in understanding the temporal relationship of multiple audios by contrastively post–training an ALM with augmented dataset. We prove our claims with rigorous experiments across benchmark tasks such as Retrieval and Classification, and ZS evaluation tasks such as ZSTE.

---

> ### Author Response · Authors · 2024-11-28
> **Limitations of our work**
>
> We humbly accept the suggestions by the reviewers and have made a full list of our model's limitation. Importantly it is to be noted that all these limitations are well out of scope of our current work and hence doesn't take away any novelty from our work.
>
> > **What our work doesn't cover? A note to all future authors interested in this work:**
>
> > - **General-Purpose ALM:** Our proposed model is not a general-purpose Audio-Language Model (ALM) capable of performing all downstream applications across all datasets. The current implementation is specifically designed and validated on the ESC-50 dataset as a proof-of-concept and to achieve our defined objectives. Consequently, generalization remains a limitation, although addressing this was beyond the scope of our work.
>  > - **Temporality Beyond the Dataset:** The model does not provide a general understanding of temporality beyond the ESC-50 dataset. Results from the ZSTE Task 4 and 5 confirm that neither does the model propose general temporality, nor does it achieve it. Notably, we emphasize that achieving general-purpose temporality would require larger, more comprehensive pre-trained text encoders. Specifically, open-ended text encoders (e.g., encoders from encoder-decoder models) would be more suitable than encoders trained on closed masked language modeling techniques, such as BERT.
> > - **Zero-Shot Evaluation Scheme:** While our zero-shot evaluation scheme is designed to be general-purpose, it is inherently limited to contrastive models. Furthermore, the evaluation has not been tested on domains beyond the ESC-50 dataset. The reported ZSTE results are restricted to this dataset because it aligns with the training data used in our model and those of prior works. To mitigate data leakage, we ensure that our evaluation dataset is separate from the training data. However, it is important to note that the primary objective of this evaluation is to validate the proof-of-concept rather than to set new benchmarks.
> > - **Evaluation Dataset Overlap:** A broader limitation, relevant to most models in this domain, is the overlap between evaluation datasets across various benchmarks. These overlaps can occur in terms of sound events or contextual similarities. Therefore, we caution against uncritical comparisons of model performance on classical benchmarks without careful consideration of dataset overlap.
>
> By addressing these limitations, we hope to guide future researchers in expanding and improving upon the current work to achieve broader applicability and more generalized performance.

---

> ### Author Response · Authors · 2024-11-28
> **Request to update the score towards acceptance of the work.**
>
> **Final Note:**
>
> > As the updation of the drafts reach its final day, we have rigoursly went through each reviewer's comment and tried to address it systematically. **If the final version of the paper and our supported responses address your concerns and questions, then we kindly request an updated score** to reflect the novelty and potential impact of this work in advancing the field of audio-language models and multi-modality. Thank you.

---

> > ### Author Response · Authors · 2024-12-03
> > **Final request towards reviewers (2nd Dec)**
> >
> > Dear Reviewers and Chairs,
> >
> > We would like to ask the Chairs to kindly notify the reviewers as well. As the review window approaches its conclusion tonight **(2nd Dec 11:59 pm AoE)**,
> >
> > It's our final request to kindly review our rebuttal responses and adequately address your concerns. If you find that all your concerns have been addressed, we would humbly request you to consider updating your scores. **We have tried our best to address all the concerns methodologically and also expicitly mention the limitations of our work**. Kindly reflect to these changes and reward the paper a better score.
> >
> > Thank you.
> >
> > --Authors.

---

### Meta-Review · Area_Chair_LtVd · 2024-12-22

**Metareview:**

This paper presents a post-training method to improve the CLAP model's understanding of temporal relations in audio events (such as before/after). The approach involves a two-stage training strategy: 1) post-training on single and dual audio events, and 2) post-training on varying temporal sequences with the same event combinations.

There are three main contributions claimed by authors: 1) The analysis showing the current contrastive Audio-Language Models (ALMs) failed to capture temporal audio relationships; 2) proposed two-stage post training method TeminAL; 3) ZSTE: zero-shot temporal evaluation scheme.

Regarding the first contribution, the reviewers have pointed out that multiple literatures already had the same observation. Regarding the second contribution, there are also multiple methods have already explored this direction such as Comp-A. More importantly, the generation is a concern. There are various audio temporal events while the paper already studied a very narrow one: before/after/while. Furthermore, there may be more than two events in an audio, the authors didn’t show how to generalize to 3+ events. Another concern is about the fairness of the experiments. The authors leveraged a pretrained model which was trained with larger amounts of data. Therefore, the model performance comparison may not be apple-to-apple. Some reviewers also asked about the rationale behind using ESC50 and potential overfitting. The authors kept arguing it is beyond the scope and budget of the current work, but this seems to be not a valid excuse.

Overall, while this paper addresses a meaningful topic, its scope is too narrow and lacks generalization. Additionally, its novelty is limited due to existing similar works.

Note the revision has half-page technical content on page 11, which implies desk rejection.

In summary, the strength of this paper is that it proposed  a post-training method to improve the CLAP model's understanding of temporal relations in audio events. The primary weakness of the paper is that its contribution lacks significance. This is particularly evident given the narrow focus on temporal events (such as before/after), and the ambiguity regarding whether the study's findings can be generalized to other audio temporal events.

**Additional Comments On Reviewer Discussion:**

The initial scores of this paper are consistent with 3, 3, 3, 5. In the rebuttal, the authors actively addressed reviewers’ comments. The paper quality has clearly improved, with the final scores as 3, 5, 5, 6.

The authors have requested reviewers to update scores too many times, including setting daily reminders in multiple stages. While some reviewers have responded during rebuttal, too frequent reminders from the authors might not be necessary.

Note that the authors may be optimizing too much toward paper acceptance by even changing the claim. As pointed by reviewer YCxA, the authors just added the statement  "In contrast, our approach achieves significant advancements within a limited computational budget, training around 10% of the total trainable parameters and utilizing a single dataset (ESC-50)." in the revision while it was not presented in the initial submission.

---

### Decision · Program_Chairs · 2025-01-22

Reject